# Temporally Disentangled Representation Learning under Unknown Nonstationarity

**Xiangchen Song**[1*]   **Weiran Yao**[2]   **Yewen Fan**[1]   **Xinshuai Dong**[1]   **Guangyi Chen**[1,3]
**Juan Carlos Niebles**[2]   **Eric Xing**[1,3]   **Kun Zhang**[1,3]
[1]Carnegie Mellon University
[2]Salesforce Research
[3]Mohamed bin Zayed University of Artificial Intelligence

## Abstract

In unsupervised causal representation learning for sequential data with time-delayed latent causal influences, strong identifiability results for the disentanglement of causally-related latent variables have been established in stationary settings by leveraging temporal structure. However, in *nonstationary* setting, existing work only partially addressed the problem by either utilizing observed auxiliary variables (e.g., class labels and/or domain indexes) as side-information or assuming simplified latent causal dynamics. Both constrain the method to a limited range of scenarios. In this study, we further explored the Markov Assumption under time-delayed causally related process in *nonstationary* setting and showed that under mild conditions, the independent latent components can be recovered from their nonlinear mixture up to a permutation and a component-wise transformation, *without* the observation of auxiliary variables. We then introduce NCTRL, a principled estimation framework, to reconstruct time-delayed latent causal variables and identify their relations from measured sequential data only. Empirical evaluations demonstrated the reliable identification of time-delayed latent causal influences, with our methodology substantially outperforming existing baselines that fail to exploit the nonstationarity adequately and then, consequently, cannot distinguish distribution shifts.

## 1 Introduction

Causal reasoning for time-series data is a long-lasting yet fundamental task [1–3]. The majority of the studies focus on the temporal causal discovery among observed variables [4–6]. However, in many real-world scenarios, the observed data (e.g., image pixels in videos) instead of having direct causal edges, are generated by some causally related latent temporal processes or confounders. Learning causal relations has practical use cases, which benefit a lot of downstream tasks. However, estimating latent causal structures among those unobserved variables purely from observations without appropriate class of assumptions is an extremely challenging task (i.e. the latent variables are generally not identifiable) [7, 8].

Under the topic of unsupervised representation learning via nonlinear Independent Component Analysis (ICA), some strong identifiability results of the latent variables have been established [9–14] by introducing side information such as class labels and domain indices. Specifically focusing on time-series data, history information is also widely used as the side information for the identifiability of latent processes [15–18]. However, existing studies mainly focused on and derived identifiability results in stationary settings [10, 16] (Fig 1 (a)) or nonstationary settings with explicitly observed domain indices [12, 17, 18] (Fig 1 (b)).

---

*Part of the work was done while interning at Salesforce Research

37th Conference on Neural Information Processing Systems (NeurIPS 2023).

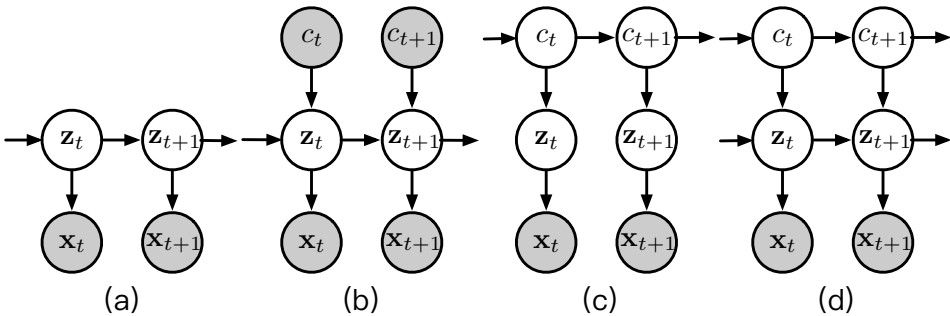

(a)        (b)        (c)        (d)

Figure 1: Graphic models for three different settings in causally related time-delayed time series data with a visual illustration. (a) is a *stationary* setting in which the transition function $\mathbf{z}_{t+1} = f_z(\mathbf{z}_t)$ stays universally the same. (b) is the setting widely explored in existing work, in which the transition function $f_z$ changes according to different domains (denoted as $c_t$), and all those domain indices are observed. (c) capture the unobserved domain indices by introducing a Markov chain on $c_t$. (d) is a more general form to model the time series data in this work. It allows nonstationary settings and it doesn't require the domain indices to be observed.

One can immediately tell the infeasibility of those two scenarios that general time-series data is usually nonstationary and the side information (class labels and domain indices) is usually unobserved. That is particularly true when considering real-world data such as video or signal sequences. It doesn't make any sense to assume that there exists a stationary transition function that is applied to the whole video clip. Take a very simple video clip of a mouse[2] [19] as an example, it is fairly clear that such a simple motion example can be divided into at least two phases (1) active phase in which the mouse is moving and (2) inactive phase in which the mouse is laying down. Instead of using a complex transition function to describe the whole video clip, a more reasonable assumption is that the same transition function is shared within the same phase, but across different phases, the transition functions are different, in other words, the transition function can be expressed as a function of the domain index. Also, it is worth mentioning that if such domain or phase indices is latent or unobserved, then we cannot directly utilize the existing framework to learn the latent causal dynamics. That is again a more realistic case that in general, the domain indices within a video are not accessible without expensive human annotation.

Recently, HMNLICA [14] attempted to resolve the problem by introducing Markov Assumption on the nonstationary discrete domain variable, they assumed the domain indices follow a first-order Markov Chain and estimated the domain information purely from observed data. However, HMN-LICA assumes temporally mutually independent sources in the data-generating process (conditioning on domain indices), i.e. they don't allow latent variables to have time-delayed causal relations in between (Fig 1 (c)). Such an assumption imposed a huge negative impact on the usability of those methods. Considering the video of the little mouse example, the $\mathbf{x}_t$s are the observed video frames, $\mathbf{z}_t$s can be the independent motion dynamics or causal process such as position, velocity, (angular) momentum, etc, and $c_t$s are the phases or actions such as standing up (active) and laying down (inactive). To accommodate for such general sequential data, time-delayed temporal dependence should be considered in the latent $\mathbf{z}_t$ space (Fig 1 (d)), otherwise, it is impossible to model a complex video data's temporal relation purely from discrete, domain indices. Also to make sure that the latent independent components can be recovered, temporally conditional independence should also be enforced, i.e. Each dimension of $\mathbf{z}_t$ is conditionally independent given the history $\mathbf{z}_{\text{history}}$. To this end, a natural question is:

> ***How can we establish identifiability of nonlinear ICA for general sequential data with nonstationary causally-related process without observing auxiliary variable?***

To answer this question, we first formulate the latent nonstationary states as a discrete Markov process and further explore the Markov Assumption [20] which is introduced for identifiability of nonlinear ICA in HMNLICA [14] and provided stronger identifiability result corresponding to the conditional emission distribution (i.e. the transition function of different domains) and the transition matrix of

---

[2]`https://dattalab.github.io/moseq2-website/images/sample-extraction.gif`.

the Markov process. Specifically, we generalized the identifiability of Hidden Markov Models in [20] to accommodate time-delayed causally-related non-parametric transitions in latent space (Thm. 1). Then we utilize the linear independence (Thm. 2) to further establish the identifiability of $\mathbf{z}_t$.

The main contributions of this work can be summarized as follows:

- To our best knowledge, this is the first identifiability result that can handle the nonstationary time-delayed causally-related latent temporal processes without the auxiliary variable. We formulate the problem, especially the nonstationary states into the Markov process, establish identifiability purely from observed data, and then show strong identifiability of latent independent components.

- We present NCTRL, Nonstationary Causal Temporal Representation Learning, a principled framework to recover time-delayed latent causal variables and identify their relations from measured sequential data under unobserved different distribution shifts.

- Experiments on both synthetic and real-world datasets demonstrate the effectiveness of the proposed method in recovering the latent variables.

## 2 Problem Formulation

### 2.1 Time Series Generative Model

Assume we observe $n$-dimensional time-series data at discrete time steps, $\mathbf{X} = \{\mathbf{x}_1, \mathbf{x}_2, \dots, \mathbf{x}_T\}$, where each $\mathbf{x}_t \in \mathcal{X}$ is generated from time-delayed causally related hidden components $\mathbf{z}_t \in \mathbb{R}^n$ by the invertible mixing function:

$$\mathbf{x}_t = \mathbf{g}(\mathbf{z}_t). \tag{1}$$

In addition to latent components $\mathbf{z}_t$, there is an extra hidden variable $c_t$ which is discrete with cardinality $|c_t| = C$, it follows a first-order Markov process controlled by a $C \times C$ transition matrix $\mathbf{A}$, in which the $i, j$-th entry $A_{i,j}$ is the probability to transit from state $i$ to $j$.

$$c_1, c_2, \dots, c_t \sim \text{Markov Chain}(\mathbf{A}) \tag{2}$$

For $i \in \{1, \dots, n\}$, $z_{it}$, as the $i$-th component of $\mathbf{z}_t$, is generated by (some) components of history information $\mathbf{z}_{t-1}$, discrete nonstationary indicator $c_t$, and noise $\epsilon_{it}$.

$$z_{it} = f_i(\{z_{j,t-1} \mid z_{j,t-\tau} \in \mathbf{Pa}(z_{it})\}, c_t, \epsilon_{it}) \quad with \quad \epsilon_{it} \sim p_{\epsilon_i} \tag{3}$$

where $\mathbf{Pa}(z_{it})$ is the set of latent factors that directly cause $z_{it}$, which can be any subset of $\mathbf{z}_{\text{Hx}} = \{\mathbf{z}_{t-1}, \mathbf{z}_{t-2}, \dots, \mathbf{z}_{t-L}\}$ up to history information maximum lag $L$. The components of $\mathbf{z}_t$ are mutually independent conditional on $\mathbf{z}_{\text{Hx}}$ and $c_t$.

### 2.2 Identifiability of Latent Causal Processes and Time-Delayed Latent Causal Relations

We define the identifiability of time-delayed latent causal processes in the representation function space in **Definition 1**. Furthermore, if the estimated latent processes can be identified at least up to permutation and component-wise invertible nonlinearities, the latent causal relations are also immediately identifiable because conditional independence relations fully characterize time-delayed causal relations in a time-delayed causally sufficient system, in which there are no latent causal confounders in the (latent) causal processes. Note that invertible component-wise transformations on latent causal processes do not change their conditional independence relations.

**Definition 1** (Identifiable Latent Causal Processes). *Formally let $\mathbf{X} = \{\mathbf{x}_1, \mathbf{x}_2, \dots, \mathbf{x}_T\}$ be a sequence of observed variables generated by the true temporally causal latent processes specified by $(f_i, p(\epsilon_i), \mathbf{A}, \mathbf{g})$ given in Eqs. (1), (2), and (3). A learned generative model $(\hat{f}_i, \hat{p}(\epsilon_i), \hat{\mathbf{A}}, \hat{\mathbf{g}})$ is observationally equivalent to $(f_i, p(\epsilon_i), \mathbf{A}, \mathbf{g})$ if the model distribution $p_{\hat{g}, \hat{p}_\epsilon, \hat{\mathbf{A}}, \hat{\mathbf{g}}}(\{\mathbf{x}_1, \mathbf{x}_2, \dots, \mathbf{x}_T\})$ matches the data distribution $p_{f_i, p_\epsilon, \mathbf{A}, \mathbf{g}}(\{\mathbf{x}_1, \mathbf{x}_2, \dots, \mathbf{x}_T\})$ everywhere. We say latent causal processes are identifiable if observational equivalence can lead to identifiability of the latent variables up to permutation $\pi$ and component-wise invertible transformation $T$:*

$$p_{\hat{f}_i, \hat{p}_{\epsilon_i}, \hat{\mathbf{A}}, \hat{\mathbf{g}}}(\{\mathbf{x}_1, \mathbf{x}_2, \dots, \mathbf{x}_T\}) = p_{f_i, p_{\epsilon_i}, \mathbf{A}, \mathbf{g}}(\{\mathbf{x}_1, \mathbf{x}_2, \dots, \mathbf{x}_T\})$$
$$\Rightarrow \hat{\mathbf{g}}^{-1}(\mathbf{x}_t) = T \circ \pi \circ \mathbf{g}^{-1}(\mathbf{x}_t), \quad \forall \mathbf{x}_t \in \mathcal{X}, \tag{4}$$

*where $\mathcal{X}$ is the observation space.*

# 3 Identifiability Theory

In this section, we showed that under mild conditions, the latent variable $\mathbf{z}_t$ is identifiable up to permutation and a component-wise transformation. The theoretical results can be divided into two parts (1) identifiability of the nonstationarity and (2) identifiability of the independent components. As introduced above, the major challenge comes from the unobserved domain indices or nonstationary indicators ($c_t$ in our graphic models). We first establish the identifiability of the different conditional distributions from the observed data and then show that the latent variables $\mathbf{z}$ are identifiable. The complete proofs can be found in Appendix A.

## 3.1 Identifiability of Nonstationary Hidden States

Gassiat et al.[20] showed that the conditional emission distributions in Hidden Markov Models and the transition matrix are identifiable up to label swapping. We first generalize it to the autoregressive setting to accommodate for the time-delayed causal relation, i.e. we showed the identifiability of conditional emission distributions $p(\mathbf{x}_t|\mathbf{x}_{t-1}, c)$.

**Theorem 1.** *(identifiability of the nonstationarity with Markov Assumptions) Suppose the observed data is generated following the nonlinear ICA framework as defined in Eqs.* (1)*,* (2) *and* (3)*. Suppose the following assumptions (Markov Assumptions) hold:*

> *i For the Markov process, the number of latent states, $C$, is known.*

> *ii The transition matrix $\mathbf{A}$ is full rank.*

*Use $\mu_1, \ldots, \mu_C \in \mathbb{R}^n$ to denote nonparametric probability distributions of the $C$ emission distributions $\mu_c = p(\mathbf{x}_t \mid \mathbf{x}_{t-1}, c)$. Then the parameters $\mathbf{A}$ and $M = (\mu_1, \ldots, \mu_C)$ are identifiable given the distribution, $\mathbb{P}_{\mathbf{A},M}^{(4)}$, of at least 4 consecutive observations $\mathbf{x}_t, \mathbf{x}_{t+1}, \mathbf{x}_{t+2}, \mathbf{x}_{t+3}$, up to label swapping of the hidden states, that is:*

*If $\widetilde{\mathbf{A}}$ is a $C \times C$ transition matrix and if $\widetilde{\pi}(c)$ is a stationary distribution of $\widetilde{\mathbf{A}}$ with $\widetilde{\pi}(c) > 0$ $\forall c \in \{1, \ldots, C\}$, and if $\tilde{M} = (\tilde{\mu}_1, \ldots, \tilde{\mu}_C)$ are $C$ probability distributions on $\mathbb{R}^n$ that verify the equality of the distribution functions $\mathbb{P}_{\widetilde{\mathbf{A}}, \tilde{M}}^{(4)} = \mathbb{P}_{\mathbf{A},M}^{(4)}$, then there exists a permutation $\sigma$ of the set $\{1, \ldots, C\}$ such that for all $k, l = 1, \ldots, C$ we have $\tilde{A}_{k,l} = A_{\sigma(k), \sigma(l)}$ and $\tilde{\mu}_k = \mu_{\sigma(k)}$.*

For notational simplicity, and without loss of generality, we can assume the components are ordered such that $c = \sigma(c)$. That leads us to the identifiability of the nonstationarity in the system i.e. up to label swapping of the hidden states, the conditional emission distributions $p(\mathbf{x}_t|\mathbf{x}_{t-1}, c_t)$ and transition matrix $\mathbf{A}$ are identifiable, hence providing us a bridge to further leverage the temporal independence condition in the latent space to establish the identifiability result for demixing function or in other words the latent variables $\mathbf{z}_t$.

## 3.2 Identifiability of Latent Causal Processes

To incorporate nonlinear ICA into the Markov Assumption we define the emission distribution $p(\mathbf{x}_t \mid \mathbf{x}_{t-1}, c)$ as a deep latent variable model. First, the latent independent component variables $\mathbf{z}_t \in \mathbb{R}^n$ are generated from a factorial prior, given the hidden state $c_t$ and previous $\mathbf{z}_{t-1}$, as

$$p(\mathbf{z}_t \mid \mathbf{z}_{t-1}, c_t) = \prod_{k=1}^{n} p(z_{kt} \mid \mathbf{z}_{t-1}, c_t). \tag{5}$$

Second, the observed data $\mathbf{x}_t$ is generated by a nonlinear mixing function as in Eq. (1) which is assumed to be bijective with inverse given by $\mathbf{z}_t = \mathbf{g}(\mathbf{x}_t)$. Let $\eta_{kt}(c_t) \triangleq \log p(z_{kt}|\mathbf{z}_{t-1}, c_t)$, and assume that $\eta_{kt}(c_t)$ is twice differentiable in $z_{kt}$ and is differentiable in $z_{l,t-1}$, $l = 1, 2, ..., n$. Note that the parents of $z_{kt}$ may be only $c_t$ and a subset of $\mathbf{z}_{t-1}$; if $z_{l,t-1}$ is not a parent of $z_{kt}$, then $\frac{\partial \eta_{lk}}{\partial z_{l,t-1}} = 0$.

**Theorem 2.** *(identifiability of the independent components) Suppose there exists an invertible function $\hat{\mathbf{g}}^{-1}$, which is the estimated demixing function that maps $\mathbf{x}_t$ to $\hat{\mathbf{z}}_t$, i.e.,*

$$\hat{\mathbf{z}}_t = \hat{\mathbf{g}}^{-1}(\mathbf{x}_t) \tag{6}$$

*such that the components of $\hat{\mathbf{z}}_t$ are mutually independent conditional on $\hat{\mathbf{z}}_{t-1}$. Let*

$$\mathbf{v}_{k,t}(c) \triangleq \left( \frac{\partial^2 \eta_{kt}(c)}{\partial z_{k,t} \partial z_{1,t-1}}, \frac{\partial^2 \eta_{kt}(c)}{\partial z_{k,t} \partial z_{2,t-1}}, ...., \frac{\partial^2 \eta_{kt}(c)}{\partial z_{k,t} \partial z_{n,t-1}} \right)^{\mathsf{T}},$$

$$\mathring{\mathbf{v}}_{k,t}(c) \triangleq \left( \frac{\partial^3 \eta_{kt}(c)}{\partial z_{k,t}^2 \partial z_{1,t-1}}, \frac{\partial^3 \eta_{kt}(c)}{\partial z_{k,t}^2 \partial z_{2,t-1}}, ...., \frac{\partial^3 \eta_{kt}(c)}{\partial z_{k,t}^2 \partial z_{n,t-1}} \right)^{\mathsf{T}}. \tag{7}$$

*And*

$$\mathbf{s}_{kt} \triangleq \left( \mathbf{v}_{kt}(1)^{\mathsf{T}}, ..., \mathbf{v}_{kt}(C)^{\mathsf{T}}, \frac{\partial^2 \eta_{kt}(2)}{\partial z_{kt}^2} - \frac{\partial^2 \eta_{kt}(1)}{\partial z_{kt}^2}, ..., \frac{\partial^2 \eta_{kt}(C)}{\partial z_{kt}^2} - \frac{\partial^2 \eta_{kt}(C-1)}{\partial z_{kt}^2} \right)^{\mathsf{T}},$$

$$\mathring{\mathbf{s}}_{kt} \triangleq \left( \mathring{\mathbf{v}}_{kt}(1)^{\mathsf{T}}, ..., \mathring{\mathbf{v}}_{kt}(C)^{\mathsf{T}}, \frac{\partial \eta_{kt}(2)}{\partial z_{kt}} - \frac{\partial \eta_{kt}(1)}{\partial z_{kt}}, ..., \frac{\partial \eta_{kt}(C)}{\partial z_{kt}} - \frac{\partial \eta_{kt}(C-1)}{\partial z_{kt}} \right)^{\mathsf{T}}. \tag{8}$$

*If for each value of $\mathbf{z}_t$, $\mathbf{s}_{1t}, \mathring{\mathbf{s}}_{1t}, \mathbf{v}_{2t}, \mathring{\mathbf{s}}_{2t}, ..., \mathbf{s}_{nt}, \mathring{\mathbf{s}}_{nt}$, as $2n$ function vectors $\mathbf{s}_{k,t}$ and $\mathring{\mathbf{s}}_{k,t}$, with $k = 1, 2, ..., n$, are linearly independent, then $\hat{\mathbf{z}}_t$ must be an invertible, component-wise transformation of a permuted version of $\mathbf{z}_t$.*

So far, the identifiability result has been established without observing the nonstationarity indicators such as domain indices. In the next section, a novel Variational Auto-Encoder based method is introduced to estimate the demixing function $\hat{\mathbf{g}}^{-1}$.

## 4 NCTRL: **Nonstationary Causal Temporal Representation Learning**

In this section, we present the details of NCTRL to estimate the latent causal processes under unobserved nonstationary distribution shift, given the identifiability results in Sec 3. First, we show that our framework includes three modules, Autoregressive Hidden Markov Module, Prior Network, and Encoder-Decoder Module. Then, we provide the optimization objective of our model training including an HMM free energy lower bound, a reconstruction likelihood loss, and a KL divergence.

### 4.1 Model Architecture

Our framework extends Sequential Variational Auto-Encoders [21] with tailored modules to model nonstationarity, and enforces the conditions in Sec. 3 as constraints. We give the estimation procedure of the latent causal dynamics model in Eq. (3). The model architecture is showcased in Fig. 2. The framework has three major components (1) Autoregressive Hidden Markov Module (ARHMM), (2) Prior Network Module, and (3) Encoder-Decoder Module.

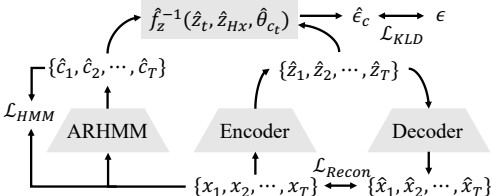

Figure 2: Illustration of NCTRL with (1) Autoregressive Hidden Markov Module, (2) Prior Network, and (3) Encoder-Decoder Module.

**Autoregressive Hidden Markov Module (ARHMM)**    The first component of our framework is ARHMM which deals with the nonstationarity with unobserved domains. As discussed in Thm 1, the transition function or the conditional emission distributions across different domains together with the Markov transition matrix $\mathbf{A}$ are identifiable. This module estimates the transition function of different domains $p(\mathbf{x}_t | \mathbf{x}_{t-1}, c_t)$ and the transition matrix $\mathbf{A}$ of the Markov process, and ultimately decodes the optimal domain indices $\{\hat{c}_1, \hat{c}_2, \ldots, \hat{c}_T\}$ via the Viterbi algorithm.

**Prior Network Module**    To better estimate the prior distribution $p(\hat{z}_t | \hat{\mathbf{z}}_{\mathrm{Hx}}, c_t)$, let $\mathbf{z}_{\mathrm{Hx}}$ denote the lagged latent variables up to maximum time lag $L$. We evaluate $p(\hat{z}_t | \hat{\mathbf{z}}_{\mathrm{Hx}}, c_t) = p_\epsilon \left( \hat{f}_z^{-1}(\hat{z}_t, \hat{\mathbf{z}}_{\mathrm{Hx}}, \hat{\boldsymbol{\theta}}_{c_t}) \right) \left| \frac{\partial \hat{f}_z^{-1}}{\partial \hat{z}_t} \right|$ by learning a holistic inverse dynamics $\hat{f}_z^{-1}$ that takes the estimated change factors for dynamics $\hat{\boldsymbol{\theta}}_{c_t}$ as inputs. Conditional independence of the estimated latent variables $p(\hat{\mathbf{z}}_t | \hat{\mathbf{z}}_{\mathrm{Hx}})$ is enforced by summing up all estimated component densities when obtaining the joint $p(\mathbf{z}_t | \mathbf{z}_{\mathrm{Hx}}, c_t)$ in Eq. 9. Given that the Jacobian is lower-triangular, we can compute its determinant as

the product of diagonal terms. The detailed derivations are given in Appendix B.2.

$$\log p\left(\hat{\mathbf{z}}_t | \hat{\mathbf{z}}_{\mathrm{Hx}}, c_t\right) = \underbrace{\sum_{i=1}^{n} \log p(\hat{\epsilon}_i | c_t)}_{\text{Conditional indepdence}} + \underbrace{\sum_{i=1}^{n} \log \left| \frac{\partial \hat{f}_i^{-1}}{\partial \hat{z}_{it}} \right|}_{\text{Lower-triangular Jacobian}} \tag{9}$$

**Encoder-Decoder Module** The third component is a Variational Auto-Encoder based module which utilizes reconstruction loss to enforce the invertibility of learned mixing function $\hat{\mathbf{g}}$. Specifically, the encoder fits the demixing function $\hat{\mathbf{g}}^{-1}$ and the decoder fits the mixing function $\hat{\mathbf{g}}$. The implementation details are in Appendix B.

### 4.2 Optimization

The first training objective of NCTRL is to maximize the Log-likelihood of the observed data:

$$\log p_{\boldsymbol{\theta}_{\mathrm{HMM}}}(\{\mathbf{x}_1, \mathbf{x}_2, \dots, \mathbf{x}_T\}) \tag{10}$$

where $\boldsymbol{\theta}_{\mathrm{HMM}}$ represents the HMM training parameters. Then the free energy lower bound can be defined as:

$$-\mathcal{L}_{\mathrm{HMM}} = \mathcal{L}(q(\mathbf{c}), \boldsymbol{\theta}_{\mathrm{HMM}}) \triangleq \mathbb{E}_{q(\mathbf{c})}\left[\log p_{\boldsymbol{\theta}_{\mathrm{HMM}}}(\mathbf{x}_1, \mathbf{x}_2, \dots, \mathbf{x}_T, \mathbf{c})\right] - \mathbf{H}(q(\mathbf{c})) \tag{11}$$

Consistent with the theory part, the first training objective is to maximize data log-likelihood in the ARHMM module to get optimal $q(\mathbf{c}^{\star})$.

$$q(\mathbf{c}^{\star}) \triangleq \arg\max_{q(\mathbf{c})} \mathcal{L}(q(\mathbf{c}), \boldsymbol{\theta}_{\mathrm{HMM}}) \tag{12}$$

which can easily be computed by the Forward-Backward algorithm and luckily all of it is differentiable to the HMM training parameters $\boldsymbol{\theta}_{\mathrm{HMM}}$(transition matrix $\mathbf{A}$ and transition function parameters $\boldsymbol{\theta}_f$).

Then the second part is to maximize the Evidence Lower BOund (ELBO) for the VAE framework, which can be written as (complete derivation steps are in Appendix B.3):

$$\begin{aligned}
\mathrm{ELBO} &\triangleq \log p_{\mathrm{data}}(\mathbf{X}) - D_{KL}(q_\phi(\mathbf{Z}|\mathbf{X}) || p_{\mathrm{data}}(\mathbf{Z}|\mathbf{X})) \\
&= \underbrace{\mathbb{E}_{\mathbf{z}_t} \sum_{t=1}^{T} \log p_{\mathrm{data}}(\mathbf{x}_t | \mathbf{z}_t)}_{-\mathcal{L}_{\mathrm{Recon}}} + \underbrace{\mathbb{E}_{\mathbf{c}}\left[ \sum_{t=1}^{T} \log p_{\mathrm{data}}(\mathbf{z}_t | \mathbf{z}_{\mathrm{Hx}}, c_t) - \sum_{t=1}^{T} \log q_\phi(\mathbf{z}_t | \mathbf{x}_t) \right]}_{-\mathcal{L}_{\mathrm{KLD}}}
\end{aligned} \tag{13}$$

We use mean-squared error (MSE) for the reconstruction likelihood loss $\mathcal{L}_{\mathrm{Recon}}$. The KL divergence $\mathcal{L}_{\mathrm{KLD}}$ is estimated via a sampling approach since with a learned nonparametric transition prior, the distribution does not have an explicit form. Specifically, we obtain the log-likelihood of the posterior, evaluate the prior $\log p\left(\hat{\mathbf{z}}_t | \hat{\mathbf{z}}_{\mathrm{Hx}}, c_t\right)$ in Eq. (9), and compute their mean difference in the dataset as the KL loss: $\mathcal{L}_{\mathrm{KLD}} = \mathbb{E}_{\hat{\mathbf{z}}_t \sim q(\hat{\mathbf{z}}_t | \mathbf{x}_t)} \log q(\hat{\mathbf{z}}_t | \mathbf{x}_t) - \log p\left(\hat{\mathbf{z}}_t | \hat{\mathbf{z}}_{\mathrm{Hx}}, c_t\right)$.

## 5 Experiments

We evaluate the identifiability results of NCTRL on a number of simulated and real-world temporal datasets. We first introduce the evaluation metrics and baselines and then discuss the datasets we used in our experiments. Lastly, we show the experiment results discuss the performance, and make comparisons.

### 5.1 Evaluation Metrics

**Mean Correlation Coefficient (MCC)** To evaluate the identifiability of the latent variables, we compute the Mean Correlation Coefficient (MCC) on the test dataset. MCC is a standard metric in the ICA literature for continuous variables which measure the identifiability of the learned latent causal processes. MCC is close to 1.0 when latent variables are identifiable up to permutation and component-wise invertible transformation in the noiseless case.

**Mean Square Error (MSE) for estimating A** As introduced in the theory, the $\mathbf{A}$ is identifiable in our setting, which means that our proposed method can provide accurate estimation for the transition matrix $\mathbf{A}$, to valid such a claim, we use mean square error (MSE) to capture the distance between the estimated $\hat{\mathbf{A}}$ and ground truth $\mathbf{A}$.

**Accuracy for estimating $c_t$** We also test the accuracy for estimating the discrete domain indices $c_t$ supplementary to the MSE for $\mathbf{A}$ since in theory, the $\mathbf{A}$ is identifiable but the $c_t$ is generally not identifiable, which is relatively easy to understand as an analogy in Hidden Markov Models, the transition matrix is identifiable but we can only "infer" the best possible discrete variables but cannot establish identifiability for it.

It is also worth mentioning that the MSE and Accuracy are influenced by the permutation, which is also true in clustering evaluation problems. Here we explored all permutations and selected the best possible assignment for evaluation.

## 5.2 Baselines

The following identifiable nonlinear ICA methods are used: (1) BetaVAE [22] which ignores both history and nonstationarity information. (2) i-VAE [12] and TCL [9] which leverage nonstationarity to establish identifiability but assume independent factors. (3) SlowVAE [16], and PCL [10] which exploit temporal constraints but assume independent sources and stationary processes. (4) TDRL [18] which assumes nonstationary, causal processes but with observed domain indices. (5) HMNLICA [14] which considers the unobserved nonstationary part in the data generation process but doesn't allow any causally related time-delayed relations.

## 5.3 Simulated Results

We generate two synthetic datasets corresponding to different complexity of the nonlinear mixing function $\mathbf{g}$. Both synthetic datasets satisfy our identifiability conditions in the theorems following the procedures in Appendix B.4. As in Table 1, NCTRL can recover the latent processes under unknown nonstationary distribution shifts with high MCCs (>0.95). The baselines that do not exploit history (i.e., BetaVAE, i-VAE, TCL), with independent source assumptions (Slow-VAE, PCL), consider limited nonstationary cases (TDRL) distort the identifiability results. The only baseline that considers the unknown nonstationarity in the domain indices (HMN-

Table 1: Experiment results of two synthetic datasets on baselines and proposed NCTRL, we run the experiments with five different random seeds and calculate the average with standard derivation. The best results are shown in **bold**.

| Method | Mean Correlation Coefficien (MCC) | | |
|---|---|---|---|
| | **Dataset A** | **Dataset B** | **Ave.** |
| BetaVAE | $44.02 \pm 3.11$ | $47.48 \pm 10.58$ | 45.75 |
| i-VAE | $89.74 \pm 3.38$ | $44.50 \pm 0.25$ | 67.12 |
| TCL | $37.12 \pm 0.60$ | $56.33 \pm 3.77$ | 46.73 |
| SlowVAE | $33.84 \pm 0.60$ | $53.92 \pm 3.56$ | 43.88 |
| PCL | $42.41 \pm 2.87$ | $63.66 \pm 2.77$ | 53.04 |
| HMNLICA | $59.82 \pm 4.94$ | $57.25 \pm 1.45$ | 58.54 |
| TDRL | $83.99 \pm 1.92$ | $72.02 \pm 2.76$ | 78.01 |
| NCTRL | $\mathbf{98.85 \pm 0.30}$ | $\mathbf{99.01 \pm 0.24}$ | **98.93** |

LICA) explored the Markov Assumption but doesn't allow a time-delayed causal process and hence suffers a poor result (MCC 0.58).

On the other hand, the difference between dataset A and dataset B is the nonlinearity in the mixing function, dataset A has a relatively simple nonlinear mixing function, on the contrary, dataset B has more complex nonlinearity. Some variability has been observed among the relative performance ranks of different baselines. For example, i-VAE showed a great discrepancy between the two datasets, which revived the weakness of capturing complex nonlinearity in the unknown nonstationary distribution shift environments. Again we also observed that our proposed method can constantly recover the latent independent components with high MCC which indicates on both datasets the model is identifiable and the estimation algorithm is highly effective. To further validate if NCTRL successfully recovered the Markov transition matrix $\mathbf{A}$ and inferred the domain indices $c_t$ with high accuracy. We further examine the accuracy for estimating nonstationary domain indices $c_t$ and the mean square error estimating the transition matrix $\mathbf{A}$. As shown in Table 2 the result is consistent with our theory in which the transition matrix $\mathbf{A}$ is identifiable and we can estimate it with

Table 2: Supplementary experiment results of two synthetic datasets on estimating domain indices $c_t$ and transition matrix $\mathbf{A}$ in NCTRL, we run the experiments with five different random seeds and calculate the average with standard derivation.

| | Unknown Nonstationary Metrics | |
|---|---|---|
| Dataset | Accuracy estimating $c_t$ | MSE estimating A |
| A | $89.96 \pm 0.24$ | $1.01 \times 10^{-3} \pm 1.67 \times 10^{-4}$ |
| B | $89.84 \pm 0.29$ | $1.08 \times 10^{-3} \pm 1.89 \times 10^{-4}$ |

high accuracy. For the nonstationary domain indices $c_t$ even though there is no identifiability result governing the estimation accuracy, it can still be inferred pretty well since it is nothing but a decoding problem in Hidden Markov Models.

## 5.4 Real-world Applications

**Video data – Modified CartPole Environment**   We evaluate NCTRL on the modified CartPole [23] video dataset and compare the performances with the baselines. Modified Cartpole is a nonlinear dynamical system with cart positions $x_t$ and pole angles $\theta_t$ as the true state variables. The dataset descriptions are in Appendix B.5. Similar to the synthetic dataset, we randomly initialize a Markov chain and roll out a series of $c_t$, and configure the CartPole environment with respect to the $c_t$. Specifically, we use five domains with different configurations of cart mass, pole mass, gravity, and noise levels. Together with the two discrete actions (i.e., left and right). By doing so, the nonstationarity is enforced, and since we can control and access all intermediate states in the system, all metrics including MCC and $c_t$ accuracy together with $\mathbf{A}$ MSE can be easily calculated. We fit NCTRL with two-dimensional causal factors. We set the latent size $n = 2$ and the lag number $L = 2$. In Fig. 3, the latent causal processes are recovered, as seen from (a) high MCC for the latent causal processes; (b) the latent factors are estimated up to component-wise transformation; and (c) the latent traversals confirm the two recovered latent variables correspond to the position and pole angle.

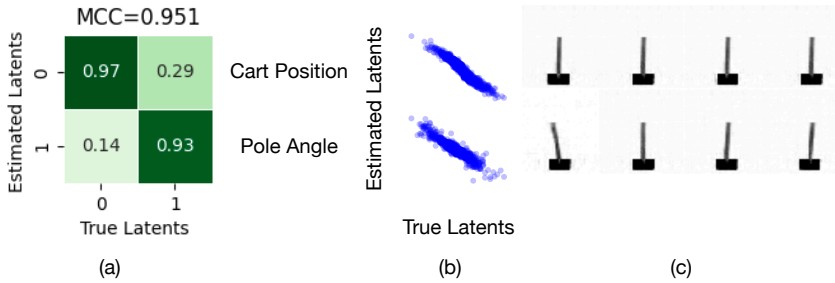

Figure 3: Modified Cartpole results: (a) MCC for causally-related factors; (b) scatterplots between estimated and true factors; and (c) latent traversal on a fixed video frame

Table 3: Experiment results of CartPole dataset. The best results are shown in **bold**.

| | | Mean Correlation Coefficient (MCC) | | | | |
|---|---|---|---|---|---|---|
| BetaVAE | i-VAE | TCL | SlowVAE | SKD | TDRL | NCTRL |
| 57.54 | 60.14 | 65.07 | 63.16 | 73.24 | 85.26 | **96.06** |

Similar to Table 1 and 2, we compare our NCTRL with baseline methods. In addition, we also compare with SKD [24], a state-of-the-art sequential disentangle representation learning method without identifiability guarantee. In Table 3 and 4 we can see that compared with TDRL, our NCTRL can recover the latent processes under unknown nonstationary distribution shifts with high MCCs (>0.95) with highly accurate estimated transition matrix $\mathbf{A}$ and high quality inferred $c_t$. Specifically, by comparing the result of SKD, the MCC for SKD is better than a variety of baselines,

Table 4: Supplementary experiment results of CartPole datasets on estimating domain indices $c_t$ and transition matrix $\mathbf{A}$ in NCTRL, we run the experiments with five different random seeds and calculate the average with standard derivation.

| Unknown Nonstationary Metrics | |
|---|---|
| **Accuracy estimating $c_t$** | **MSE estimating A** |
| $79.23 \pm 5.33$ | $5.01 \times 10^{-2} \pm 1.23 \times 10^{-2}$ |

however, we can see the distinction between well-disentangled models and identifiable models, only the models with identifiability can find the ground truth latent variables with theoretical guarantee.

**Video data – MoSeq Dataset**    We test NCTRL framework to analyze mouse behavior video data from Wiltschko et al. [19], which represents the original application to clustering mouse behavior[3], details of this dataset are available in Appendix B.6. Since there are no ground truth independent components in this particular real-world dataset, we analyze it by several visualizations to see if different domains can be properly identified and if the patterns in the recovered independent components are consistent with the recovered domain indices. We analyze the first video clip of mouse behavior data and visualize the two phases we discovered and segmented in Fig 4. We can clearly see from Fig 4 that there are different phases with the upper one actively moving and the lower one inactive. The recovered independent components showed a consistent pattern with the recovered phase or domain indices as shown in Fig 4.

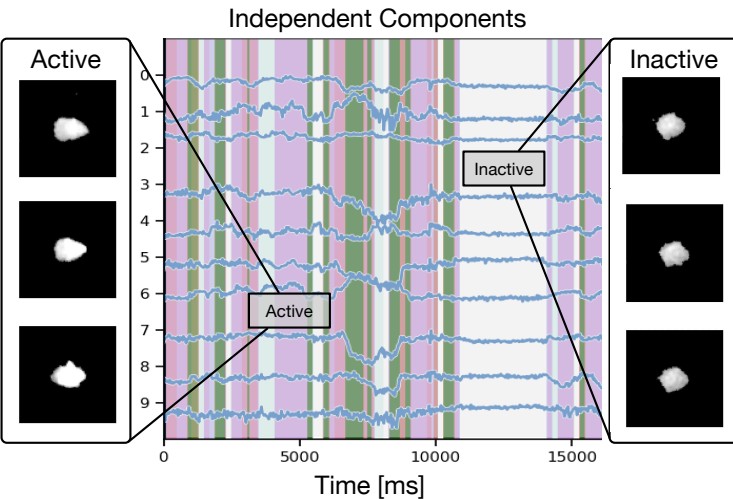

Figure 4: Result visualization of MoSeq dataset. (Active, Inactive) show two representative video frames for the active and inactive phases and (Independent Components) visualize the discovered independent components with corresponding phases tagged with different colors.

## 6    Related Work

**Causal Discovery from Time Series**    Understanding the causal structure in time-series data is pivotal in areas such as machine learning [1], econometrics [2], and neuroscience [3]. A bulk of the research in this realm emphasizes determining the temporal causal links among observed variables. The primary techniques employed are constraint-based methods [25], which use conditional independence tests to ascertain causal structures, and score-based methods [26, 27], where scores are utilized to oversee a search operation. Some researchers also proposed a combination of these two methods [28, 29]. Additionally, Granger causality [30] and its nonlinear adaptations [31, 32] have gained widespread acceptance in this context.

---

[3]Dataset can be accessed via `https://dattalab.github.io/moseq2-website/index.html`

**Nonlinear ICA for Time Series** Recently, the significance of temporal structures and non-stationarities has been recognized in achieving identifiability within nonlinear ICA. Time-contrastive learning (TCL [9]) utilizes the independent sources principle, focusing on data segments' variability. On the other hand, Permutation-based contrastive (PCL [10]) offers a learning approach that distinguishes true independent sources from shuffled ones under the uniformly dependent assumption. HMNLICA [14] integrates nonlinear ICA with an HMM to address non-stationarity without segmenting data manually. The i-VAE [12] approach employs VAEs to capture the actual joint distribution between observed and auxiliary non-stationary domains, assuming an exponential families conditional distribution. The recent advancements in nonlinear ICA for time series include LEAP [17], (i-)CITRIS [33, 34], and TDRL [18]. While LEAP introduces a novel condition emphasizing non-stationary noise, TDRL delves deeper into a non-parametric environment within a nonstationary context. In contrast, CITRIS recommends utilizing intervention target data for pinpointing latent causal aspects, avoiding certain constraints but necessitating active intervention access.

**Sequential Disentanglement** Majority of existing work about sequential disentanglement focuses on architecture based on dynamical variational autoencoder (VAE) [35]. Early works [36, 37] separate dynamic factors from static factors using probabilistic methods. Then auxiliary tasks with self-supervisory signals [38] were introduced. C-DSVAE [39] utilized contrastive penalty terms with data augmentation to introduce additional inductive biases. In R-WAE [40], Wasserstein distance was introduced to replace KL divergence. To deal with video disentanglement [41, 42] explored generative adversarial network (GAN) architectures and [43] introduced a recurrent model with adversarial loss. FAVAE, [44] proposed a factorizing VAE and [45] proposed to learn hierarchical features. Finally, SKD [24] introduced a spectral loss term that leads to structured Koopman matrices and disentanglement.

# 7 Conclusion and Discussion

**Conclusion.** In this paper, we first established an identifiability theory for general sequential data with nonstationary causally-related processes under unknown distribution shifts. Then we presented `NCTRL`, a principled framework to recover the time-delayed latent causal variable identify their causal relations from measured data, and decode high-quality domain indices under Markov assumption. Experiment results on both synthetic datasets and real-world video datasets showed that our proposed method can recover the latent causal variables and their causal relations purely from measured data with the observation of auxiliary variables or domain indices.

**Limitation.** The basic limitation of this work is that the nonstationary domain indices are assumed to follow a Markov chain. Also, this work highly relies on the latent processes to have no instantaneous causal relations but only time-delayed influences. If the resolution of the time series is much lower, then it is usually violated and one has to find a way to deal with instantaneous causal relations. Extending our theories and framework to address the scenarios when more flexibility in the domain indices transition is allowed (i.e. beyond discrete variables following a Markov chain) and to address instantaneous dependency or instantaneous causal relations will be some of our future work.

**Boarder Impacts.** This work proposes a theoretical analysis and technical methods to learn the causal representation from time-series data, which facilitate the construction of more transparent and interpretable models to understand the causal effect in the real world. This could be beneficial in a variety of sectors, including healthcare, finance, and technology. In contrast, misinterpretations of causal relationships could also have significant negative implications in these fields, which must be carefully done to avoid unfair or biased predictions.

# 8 Acknowledgment

This project has been graciously funded by NGA HM04762010002, NSF IIS1955532, NSF CNS2008248, NIGMS R01GM140467, NSF IIS2123952, NSF BCS2040381, an Amazon Research Award, NSF IIS2311990, and DARPA ECOLE HR00112390063. This project is also partially supported by NSF Grant 2229881, the National Institutes of Health (NIH) under Contract R01HL159805, a grant from Apple Inc., a grant from KDDI Research Inc., and generous gifts from Salesforce Inc., Microsoft Research, and Amazon Research.

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

*Supplement to*

## "Temporally Disentangled Representation Learning under Unknown Nonstationarity"

Appendix organization:

# A Identifiability

Assume we observe $n$-dimensional time-series data at discrete time steps, $\mathbf{X} = \{\mathbf{x}_1, \mathbf{x}_2, \ldots, \mathbf{x}_T\}$, where each $\mathbf{x}_t$ is generated from time-delayed causally related hidden components $\mathbf{z}_t \in \mathbb{R}^n$ by the invertible mixing function:

$$\mathbf{x}_t = \mathbf{g}(\mathbf{z}_t). \tag{1}$$

In addition to latent components $\mathbf{z}_t$, there is an extra hidden component $c_t$ which is a discrete variable with cardinality $|c_t| = C$, it follows first-order Markov process controlled by a $C \times C$ transition matrix $\mathbf{A}$, in which the $i, j$-th entry $A_{i,j}$ is the probability to transit from state $i$ to $j$.

$$c_1, c_2, \ldots, c_t \sim \text{Markov Chain}(\mathbf{A}) \tag{2}$$

For $i \in \{1, \ldots, n\}$, $z_{it}$, as the $i$-th component of $\mathbf{z}_t$, is generated by (some) components of history information $\mathbf{z}_{t-1}$, discrete nonstationary indicator $c_t$, and noise $\epsilon_{it}$.

$$z_{it} = f_i(\{z_{j,t-1} \mid z_{j,t-\tau} \in \mathbf{Pa}(z_{it})\}, c_t, \epsilon_{it}) \quad with \quad \epsilon_{it} \sim p_{\epsilon_i | c_t} \tag{3}$$

where $\mathbf{Pa}(z_{it})$ is the set of latent factors that directly cause $z_{it}$, which can be any subset of $\mathbf{z}_{\text{Hx}} = \{\mathbf{z}_{t-1}, \mathbf{z}_{t-2}, \ldots, \mathbf{z}_{t-L}\}$ up to history information maximum lag $L$. The components of $\mathbf{z}_t$ are mutually independent conditional on $\mathbf{z}_{\text{Hx}}$ and $c_t$.

## A.1 Identifiability of Nonstationary Hidden States

**Theorem 1.** *(identifiability of the nonstationarity with Markov Assumptions) Suppose the observed data is generated following the nonlinear ICA framework as defined in Eqs.* (1), (2) *and* (3)*. And Suppose the following assumptions (Markov Assumptions) hold:*

> *i  For the Markov process, the number of latent states, $C$, is known.*

> *ii  The transition matrix $\mathbf{A}$ is full rank.*

*Use $\mu_1, \ldots, \mu_C \in \mathbb{R}^n$ to denote nonparametric probability distributions of the $C$ emission distributions $\mu_c = p(\mathbf{x}_t \mid \mathbf{x}_{t-1}, c)$. Then the parameters $\mathbf{A}$ and $M = (\mu_1, \ldots, \mu_C)$ are identifiable given the distribution, $\mathbb{P}^{(3)}_{\mathbf{A}, M}$, of at least 4 consecutive observations $\mathbf{x}_t, \mathbf{x}_{t+1}, \mathbf{x}_{t+2}, \mathbf{x}_{t+3}$, up to label swapping of the hidden states, that is:*

*If $\widetilde{\mathbf{A}}$ is a $C \times C$ transition matrix, if $\widetilde{\pi}(c)$ is a stationary distribution of $\widetilde{\mathbf{A}}$ with $\widetilde{\pi}(c) > 0 \; \forall c \in \{1, \ldots, C\}$, and if $\tilde{M} = (\tilde{\mu}_1, \ldots, \tilde{\mu}_C)$ are $C$ probability distributions on $\mathbb{R}^n$ that verify the equality of the distribution functions $\mathbb{P}^{(3)}_{\widetilde{\mathbf{A}}, \tilde{M}} = \mathbb{P}^{(3)}_{\mathbf{A}, M}$, then there exists a permutation $\sigma$ of the set $\{1, \ldots, C\}$ such that for all $k, l = 1, \ldots, C$ we have $\tilde{A}_{k,l} = A_{\sigma(k), \sigma(l)}$ and $\tilde{\mu}_k = \mu_{\sigma(k)}$.*

*Proof.* Suppose we have:

$$\tilde{p}(\mathbf{x}_1, \ldots, \mathbf{x}_T) = p(\mathbf{x}_1, \ldots, \mathbf{x}_T) \tag{4}$$

where $p(\mathbf{x}_1, \ldots, \mathbf{x}_T)$ has transition matrix $\mathbf{A}$ and emission distributions $(\mu_1, \ldots, \mu_C)$, similarly for $\tilde{p}(\mathbf{x}_1, \ldots, \mathbf{x}_T)$.

We consider four consecutive observations $\mathbf{x}_0, \mathbf{x}_1, \mathbf{x}_2, \mathbf{x}_3$ and corresponding four discrete elements $c_0, c_1, c_2, c_3$.

$$
\begin{aligned}
p(\mathbf{x}_1, \mathbf{x}_2, \mathbf{x}_3 \mid \mathbf{x}_0) &= \sum_{c_1, c_2, c_3} p(c_1) p(\mathbf{x}_1 \mid \mathbf{x}_0, c_1) \cdot A_{c_1, c_2} p(\mathbf{x}_2 \mid \mathbf{x}_1, c_2) \cdot A_{c_2, c_3} p(\mathbf{x}_3 \mid \mathbf{x}_2, c_3) \\
&= \sum_{c_1, c_2} p(c_1) A_{c_1, c_2} p(\mathbf{x}_1 \mid \mathbf{x}_0, c_1) \cdot p(\mathbf{x}_2 \mid \mathbf{x}_1, c_2) \cdot \left( \sum_{c_3} A_{c_2, c_3} p(\mathbf{x}_3 \mid \mathbf{x}_2, c_3) \right) \\
&= \sum_{c_2} \left( \sum_{c_1} p(c_1) A_{c_1, c_2} p(\mathbf{x}_1 \mid \mathbf{x}_0, c_1) \right) \cdot p(\mathbf{x}_2 \mid \mathbf{x}_1, c_2) \cdot \left( \sum_{c_3} A_{c_2, c_3} p(\mathbf{x}_3 \mid \mathbf{x}_2, c_3) \right) \\
&= \sum_{c_2} \pi_{c_2} \underbrace{\left( \sum_{c_1} \frac{\pi_{c_1} A_{c_1, c_2}}{\pi_{c_2}} \mu_{c_1} \right)}_{\bar{\mu}_{c_2}} \cdot \mu_{c_2} \cdot \underbrace{\left( \sum_{c_3} A_{c_2, c_3} \mu_{c_3} \right)}_{\dot{\mu}_{c_2}}
\end{aligned}
\tag{5}
$$

where $\pi_{c_i} = p(c_i)$. Since $\mathbf{A}$ has full rank and the probability measures $\mu_1, \ldots, \mu_C$ are linearly independent, the probability measures $\{\bar{\mu}_{c_2} = \sum_{c_1} \frac{\pi_{c_1} A_{c_1,c_2}}{\pi_{c_2}} \mu_{c_1} \mid c_2 = 1, \ldots, C\}$ are linearly independent, and the probability measures $\{\mathring{\mu}_{c_2} = \sum_{c_3} A_{c_2,c_3} \mu_{c_3} \mid c_2 = 1, \ldots, C\}$ are also linearly independent. Thus, applying Theorem 9 of [46], there exists a permutation $\sigma$ of $\{1, \ldots, C\}$ such that, $\forall i \in \{1, \ldots, C\}$:

$$\tilde{\mu}_i = \mu_{\sigma(i)}$$

$$\sum_j \tilde{A}_{i,j} \tilde{\mu}_j = \sum_j A_{\sigma(i),j} \mu_j$$

This gives easily $\forall i \in \{1, \ldots, C\}$:

$$\sum_j \tilde{A}_{i,j} \mu_{\sigma(j)} = \sum_j A_{\sigma(i),\sigma(j)} \mu_{\sigma(j)}.$$

Since the conditional distributions $\mu_i$ are linearly independent, we can establish the equivalence between $\tilde{\mathbf{A}}$ and $\mathbf{A}$ via permutation $\sigma$,

$$\tilde{A}_{j,i} = A_{\sigma(j),\sigma(i)}, \tag{6}$$

then the theorem is proved. $\qquad\square$

For notational simplicity, and without loss of generality, we assume the components are ordered such that $c = \sigma(c)$. That leads us to the identifiability of the nonstationarity in the system i.e. up to label swapping of the hidden states, the conditional emission distributions $p(\mathbf{x}_t|\mathbf{x}_{t-1}, c_t)$ and transition matrix $\mathbf{A}$ are identifiable, hence providing us a bridge to further leverage the temporal independence condition in the latent space to establish the identifiability result for demixing function or in other words the latent variables $\mathbf{z}_t$.

## A.2 Identifiability of Latent Causal Processes

To incorporate nonlinear ICA into the Markov Assumption we define the emission distribution $p(\mathbf{x}_t \mid \mathbf{x}_{t-1}, c)$ as a deep latent variable model. First, the latent independent component variables $\mathbf{z}_t \in \mathbb{R}^n$ are generated from a factorial prior, given the hidden state $c_t$ and previous $\mathbf{z}_{t-1}$, as

$$p(\mathbf{z}_t \mid \mathbf{z}_{t-1}, c_t) = \prod_{k=1}^n p(z_{kt} \mid \mathbf{z}_{t-1}, c_t). \tag{7}$$

Second, the observed data $\mathbf{x}_t \in \mathbb{R}^n$ is generated by a nonlinear mixing function as in Eq. (1) which is assumed to be bijective with inverse given by $\mathbf{z}_t = \mathbf{g}^{-1}(\mathbf{x}_t)$. Let $\eta_{kt}(c_t) \triangleq \log p(z_{kt}|\mathbf{z}_{t-1}, c_t)$, and assume that $\eta_{kt}(c_t)$ is twice differentiable in $z_{kt}$ and is differentiable in $z_{l,t-1}$, $l = 1, 2, ..., n$. Note that the parents of $z_{kt}$ may be only $c_t$ and a subset of $\mathbf{z}_{t-1}$; if $z_{l,t-1}$ is not a parent of $z_{kt}$, then $\frac{\partial \eta_k}{\partial z_{l,t-1}} = 0$.

**Theorem 2.** *Suppose there exists an invertible function $\hat{\mathbf{g}}^{-1}$, which is the estimated demixing function that maps $\mathbf{x}_t$ to $\hat{\mathbf{z}}_t$, i.e.,*

$$\hat{\mathbf{z}}_t = \hat{\mathbf{g}}^{-1}(\mathbf{x}_t) \tag{8}$$

*such that the components of $\hat{\mathbf{z}}_t$ are mutually independent conditional on $\hat{\mathbf{z}}_{t-1}$. Let*

$$\mathbf{v}_{k,t}(c) \triangleq \left( \frac{\partial^2 \eta_{kt}(c)}{\partial z_{k,t} \partial z_{1,t-1}}, \frac{\partial^2 \eta_{kt}(c)}{\partial z_{k,t} \partial z_{2,t-1}}, ...., \frac{\partial^2 \eta_{kt}(c)}{\partial z_{k,t} \partial z_{n,t-1}} \right)^\mathsf{T},$$

$$\mathring{\mathbf{v}}_{k,t}(c) \triangleq \left( \frac{\partial^3 \eta_{kt}(c)}{\partial z_{k,t}^2 \partial z_{1,t-1}}, \frac{\partial^3 \eta_{kt}(c)}{\partial z_{k,t}^2 \partial z_{2,t-1}}, ...., \frac{\partial^3 \eta_{kt}(c)}{\partial z_{k,t}^2 \partial z_{n,t-1}} \right)^\mathsf{T}. \tag{9}$$

*And*

$$\mathbf{s}_{kt} \triangleq \left( \mathbf{v}_{kt}(1)^\mathsf{T}, ..., \mathbf{v}_{kt}(C)^\mathsf{T}, \frac{\partial^2 \eta_{kt}(2)}{\partial z_{kt}^2} - \frac{\partial^2 \eta_{kt}(1)}{\partial z_{kt}^2}, ..., \frac{\partial^2 \eta_{kt}(C)}{\partial z_{kt}^2} - \frac{\partial^2 \eta_{kt}(C-1)}{\partial z_{kt}^2} \right)^\mathsf{T},$$

$$\mathring{\mathbf{s}}_{kt} \triangleq \left( \mathring{\mathbf{v}}_{kt}(1)^\mathsf{T}, ..., \mathring{\mathbf{v}}_{kt}(C)^\mathsf{T}, \frac{\partial \eta_{kt}(2)}{\partial z_{kt}} - \frac{\partial \eta_{kt}(1)}{\partial z_{kt}}, ..., \frac{\partial \eta_{kt}(C)}{\partial z_{kt}} - \frac{\partial \eta_{kt}(C-1)}{\partial z_{kt}} \right)^\mathsf{T}. \tag{10}$$

*If for each value of $\mathbf{z}_t$, $\mathbf{s}_{1t}, \mathring{\mathbf{s}}_{1t}, \mathbf{v}_{2t}, \mathring{\mathbf{s}}_{2t}, ..., \mathbf{s}_{nt}, \mathring{\mathbf{s}}_{nt}$, as $2n$ function vectors $\mathbf{s}_{k,t}$ and $\mathring{\mathbf{s}}_{k,t}$, with $k = 1, 2, ..., n$, are linearly independent, then $\hat{\mathbf{z}}_t$ must be an invertible, component-wise transformation of a permuted version of $\mathbf{z}_t$.*

*Proof.* Combining (1) and (6) gives $\mathbf{z}_t = (\mathbf{g}^{-1} \circ \hat{\mathbf{g}})(\hat{\mathbf{z}}_t) = \mathbf{h}(\hat{\mathbf{z}}_t)$, where $\mathbf{h} \triangleq \mathbf{g}^{-1} \circ \hat{\mathbf{g}}$. Since both $\mathbf{g}$ and $\hat{\mathbf{g}}$ are invertible, $\mathbf{h}$ is invertible. Let $\mathbf{H}_t$ be the Jacobian matrix of the transformation $\mathbf{h}(\hat{\mathbf{z}}_t)$, and denote by $\mathbf{H}_{kit}$ its $(k, i)$th entry.

First, it is straightforward to see that if the components of $\hat{\mathbf{z}}_t$ are mutually independent conditional on previous $\hat{\mathbf{z}}_{t-1}$ and current $c_t$, then for any $i \neq j$, $\hat{z}_{it}$ and $\hat{z}_{jt}$ are conditionally independent given $\hat{\mathbf{z}}_{t-1} \cup (\hat{\mathbf{z}}_t \setminus \{\hat{z}_{it}, \hat{z}_{jt}\}) \cup \{c_t\}$. Mutual independence of the components of $\hat{\mathbf{z}}_t$ conditional on $\hat{\mathbf{z}}_{t-1}$ implies that $\hat{z}_{it}$ is independent from $\hat{\mathbf{z}}_t \setminus \{\hat{z}_{it}, \hat{z}_{jt}\}$ conditional on $\hat{\mathbf{z}}_{t-1}$ and $c_t$, i.e.,

$$p(\hat{z}_{it} \,|\, \hat{\mathbf{z}}_{t-1}, c_t) = p(\hat{z}_{it} \,|\, \hat{\mathbf{z}}_{t-1} \cup (\hat{\mathbf{z}}_t \setminus \{\hat{z}_{it}, \hat{z}_{jt}\}), c_t).$$

At the same time, it also implies $\hat{z}_{it}$ is independent from $\hat{\mathbf{z}}_t \setminus \{\hat{z}_{it}\}$ conditional on $\hat{\mathbf{z}}_{t-1}$ and $c_t$, i.e.,

$$p(\hat{z}_{it} \,|\, \hat{\mathbf{z}}_{t-1}, c_t) = p(\hat{z}_{it} \,|\, \hat{\mathbf{z}}_{t-1} \cup (\hat{\mathbf{z}}_t \setminus \{\hat{z}_{it}\}), c_t).$$

Combining the above two equations gives

$$p(\hat{z}_{it} \,|\, \hat{\mathbf{z}}_{t-1} \cup (\hat{\mathbf{z}}_t \setminus \{\hat{z}_{it}\}), c_t) = p(\hat{z}_{it} \,|\, \hat{\mathbf{z}}_{t-1} \cup (\hat{\mathbf{z}}_t \setminus \{\hat{z}_{it}, \hat{z}_{jt}\}), c_t),$$

i.e., for $i \neq j$, $\hat{z}_{it}$ and $\hat{z}_{jt}$ are conditionally independent given $\hat{\mathbf{z}}_{t-1} \cup (\hat{\mathbf{z}}_t \setminus \{\hat{z}_{it}, \hat{z}_{jt}\}) \cup \{c_t\}$.

We then make use of the fact that if $\hat{z}_{it}$ and $\hat{z}_{jt}$ are conditionally independent given $\hat{\mathbf{z}}_{t-1} \cup (\hat{\mathbf{z}}_t \setminus \{\hat{z}_{it}, \hat{z}_{jt}\}) \cup \{c_t\}$, then

$$\frac{\partial^2 \log p(\hat{\mathbf{z}}_t, \hat{\mathbf{z}}_{t-1}, c_t)}{\partial \hat{z}_{it} \partial \hat{z}_{jt}} = 0,$$

assuming the cross second-order derivative exists [47]. Since $p(\hat{\mathbf{z}}_t, \hat{\mathbf{z}}_{t-1}, c_t) = p(\hat{\mathbf{z}}_t \,|\, \hat{\mathbf{z}}_{t-1}, c_t) p(\hat{\mathbf{z}}_{t-1}, c_t)$ while $p(\hat{\mathbf{z}}_{t-1}, c_t)$ does not involve $\hat{z}_{it}$ or $\hat{z}_{jt}$, the above equality is equivalent to

$$\frac{\partial^2 \log p(\hat{\mathbf{z}}_t \,|\, \hat{\mathbf{z}}_{t-1}, c_t)}{\partial \hat{z}_{it} \partial \hat{z}_{jt}} = 0. \tag{11}$$

Then for any $c_t$, the Jacobian matrix of the mapping from $(\mathbf{x}_{t-1}, \hat{\mathbf{z}}_t)$ to $(\mathbf{x}_{t-1}, \mathbf{z}_t)$ is $\begin{bmatrix} \mathbf{I} & \mathbf{0} \\ * & \mathbf{H}_t \end{bmatrix}$, where $*$ stands for a matrix, and the (absolute value of the) determinant of this Jacobian matrix is $|\mathbf{H}_t|$. Therefore $p(\hat{\mathbf{z}}_t, \mathbf{x}_{t-1}|c_t) = p(\mathbf{z}_t, \mathbf{x}_{t-1}|c_t) \cdot |\mathbf{H}_t|$. Dividing both sides of this equation by $p(\mathbf{x}_{t-1}|c_t)$ gives

$$p(\hat{\mathbf{z}}_t \,|\, \mathbf{x}_{t-1}, c_t) = p(\mathbf{z}_t \,|\, \mathbf{x}_{t-1}, c_t) \cdot |\mathbf{H}_t|. \tag{12}$$

Since $p(\mathbf{z}_t \,|\, \mathbf{z}_{t-1}, c_t) = p(\mathbf{z}_t \,|\, \mathbf{g}(\mathbf{z}_{t-1}), c_t) = p(\mathbf{z}_t \,|\, \mathbf{x}_{t-1}, c_t)$ and similarly $p(\hat{\mathbf{z}}_t \,|\, \hat{\mathbf{z}}_{t-1}, c_t) = p(\hat{\mathbf{z}}_t \,|\, \mathbf{x}_{t-1}, c_t)$, Eq. 12 tells us

$$\log p(\hat{\mathbf{z}}_t \,|\, \hat{\mathbf{z}}_{t-1}, c_t) = \log p(\mathbf{z}_t \,|\, \mathbf{z}_{t-1}, c_t) + \log |\mathbf{H}_t| = \sum_{k=1}^n \eta_{kt}(c_t) + \log |\mathbf{H}_t|. \tag{13}$$

Its partial derivative w.r.t. $\hat{z}_{it}$ is

$$\frac{\partial \log p(\hat{\mathbf{z}}_t \,|\, \hat{\mathbf{z}}_{t-1}, c_t)}{\partial \hat{z}_{it}} = \sum_{k=1}^n \frac{\partial \eta_{kt}(c_t)}{\partial z_{kt}} \cdot \frac{\partial z_{kt}}{\partial \hat{z}_{it}} - \frac{\partial \log |\mathbf{H}_t|}{\partial \hat{z}_{it}}$$

$$= \sum_{k=1}^n \frac{\partial \eta_{kt}(c_t)}{\partial z_{kt}} \cdot \mathbf{H}_{kit} - \frac{\partial \log |\mathbf{H}_t|}{\partial \hat{z}_{it}}.$$

Its second-order cross-derivative is

$$\frac{\partial^2 \log p(\hat{\mathbf{z}}_t \,|\, \hat{\mathbf{z}}_{t-1}, c_t)}{\partial \hat{z}_{it} \partial \hat{z}_{jt}} = \sum_{k=1}^n \left( \frac{\partial^2 \eta_{kt}(c_t)}{\partial z_{kt}^2} \cdot \mathbf{H}_{kit} \mathbf{H}_{kjt} + \frac{\partial \eta_{kt}(c_t)}{\partial z_{kt}} \cdot \frac{\partial \mathbf{H}_{kit}}{\partial \hat{z}_{jt}} \right) - \frac{\partial^2 \log |\mathbf{H}_t|}{\partial \hat{z}_{it} \partial \hat{z}_{jt}}. \tag{14}$$

The above quantity is always 0 according to Eq. (11). Therefore, for each $l = 1, 2, ..., n$ and each value $z_{l,t-1}$, its partial derivative w.r.t. $z_{l,t-1}$ is always 0. That is,

$$\frac{\partial^3 \log p(\hat{\mathbf{z}}_t \,|\, \hat{\mathbf{z}}_{t-1}, c_t)}{\partial \hat{z}_{it} \partial \hat{z}_{jt} \partial z_{l,t-1}} = \sum_{k=1}^n \left( \frac{\partial^3 \eta_{kt}(c_t)}{\partial z_{kt}^2 \partial z_{l,t-1}} \cdot \mathbf{H}_{kit} \mathbf{H}_{kjt} + \frac{\partial^2 \eta_{kt}(c_t)}{\partial z_{kt} \partial z_{l,t-1}} \cdot \frac{\partial \mathbf{H}_{kit}}{\partial \hat{z}_{jt}} \right) \equiv 0, \tag{15}$$

where we have made use of the fact that entries of $\mathbf{H}_t$ do not depend on $z_{l,t-1}$. Using different values $r$ for $c_t$ in Eq. (14) take the difference of this equation across them gives

$$\frac{\partial^2 \log p(\hat{\mathbf{z}}_t \mid \hat{\mathbf{z}}_{t-1}; r+1)}{\partial \hat{z}_{it} \partial \hat{z}_{jt}} - \frac{\partial^2 \log p(\hat{\mathbf{z}}_t \mid \hat{\mathbf{z}}_{t-1}; r)}{\partial \hat{z}_{it} \partial \hat{z}_{jt}}$$

$$= \sum_{k=1}^n \left[ \left( \frac{\partial^2 \eta_{kt}(r+1)}{\partial z_{kt}^2} - \frac{\partial^2 \eta_{kt}(r)}{\partial z_{kt}^2} \right) \cdot \mathbf{H}_{kit}\mathbf{H}_{kjt} + \left( \frac{\partial \eta_{kt}(r+1)}{\partial z_{kt}} - \frac{\partial \eta_{kt}(r)}{\partial z_{kt}} \right) \cdot \frac{\partial \mathbf{H}_{kit}}{\partial \hat{z}_{jt}} \right] \equiv 0. \quad (16)$$

If for any value of $\mathbf{z}_t$, $\mathbf{s}_{1t}, \mathring{\mathbf{s}}_{1t}, \mathbf{s}_{2t}, \mathring{\mathbf{s}}_{2t}, ..., \mathbf{s}_{nt}, \mathring{\mathbf{s}}_{nt}$ are linearly independent, to make the above equation hold true, one has to set $\mathbf{H}_{kit}\mathbf{H}_{kjt} = 0$ or $i \neq j$. That is, in each row of $\mathbf{H}_t$ there is only one non-zero entry. Since $h$ is invertible, then $\mathbf{z}_t$ must be an invertible, component-wise transformation of a permuted version of $\hat{\mathbf{z}}_t$. $\qquad\square$

So far, the identifiability result has been established without observing the nonstationarity indicators such as domain indices.

### A.3  Discussion on Assumptions in Theorem 2

This condition was initially introduced in GCL [11], namely, "sufficient variability", to extend the modulated exponential families [9] to general modulated distributions. Essentially, the condition says that the nonstationary domains $c$ must have a sufficiently complex and diverse effect on the transition distributions. In other words, if the underlying distributions are composed of relatively many domains of data, the condition generally holds true. Loosely speaking, the sufficient variability holds if the modulation of by $c$ on the conditional distribution $q(z_{it}|\mathbf{z}_{\text{Hx}}, c)$ is not too simple in the following sense:

1. Higher order of $k$ ($k > 1$) is required. If $k = 1$, the sufficient variability cannot hold;
2. The modulation impacts $\lambda_{ij}$ by $\mathbf{u}$ must be linearly independent across domains $c$. The sufficient statistics functions $q_{ij}$ cannot be all linear, i.e., we require higher-order statistics.

Further details of this example can be found in Appendix B of [11] and Appendix S1.4.1 of [18]. In summary, we need the domains denoted by $c$ to have diverse (i.e., distinct influences) and complex impacts on the underlying data generation process.

## B  Implementation Details

### B.1  Reproducibility

All experiments are done in a GPU workstation with CPU: Intel i7-13700K, GPU: NVIDIA RTX 4090, Memory: 128 GB. The code can be found via https://github.com/xiangchensong/nctrl.

### B.2  Prior Likelihood Derivation

Let us start with an illustrative example of stationary latent causal processes consisting of two time-delayed latent variables, i.e., $\mathbf{z}_t = [z_{1,t}, z_{2,t}]$ with maximum time lag $L = 1$, i.e., $z_{i,t} = f_i(\mathbf{z}_{t-1}, \epsilon_{i,t})$ with mutually independent noises. Let us write this latent process as a transformation map $\mathbf{f}$ (note that we overload the notation $f$ for transition functions and for the transformation map):

$$\begin{bmatrix} z_{1,t-1} \\ z_{2,t-1} \\ z_{1,t} \\ z_{2,t} \end{bmatrix} = \mathbf{f} \left( \begin{bmatrix} z_{1,t-1} \\ z_{2,t-1} \\ \epsilon_{1,t} \\ \epsilon_{2,t} \end{bmatrix} \right). \quad (17)$$

By applying the change of variables formula to the map $\mathbf{f}$, we can evaluate the joint distribution of the latent variables $p(z_{1,t-1}, z_{2,t-1}, z_{1,t}, z_{2,t})$ as:

$$p(z_{1,t-1}, z_{2,t-1}, z_{1,t}, z_{2,t}) = p(z_{1,t-1}, z_{2,t-1}, \epsilon_{1,t}, \epsilon_{2,t}) / |\det \mathbf{J_f}|, \quad (18)$$

where $\mathbf{J_f}$ is the Jacobian matrix of the map $\mathbf{f}$, which is naturally a low-triangular matrix:

$$\mathbf{J_f} = \begin{bmatrix} 1 & 0 & 0 & 0 \\ 0 & 1 & 0 & 0 \\ \frac{\partial z_{1,t}}{\partial z_{1,t-1}} & \frac{\partial z_{1,t}}{\partial z_{2,t-1}} & \frac{\partial z_{1,t}}{\partial \epsilon_{1,t}} & 0 \\ \frac{\partial z_{2,t}}{\partial z_{1,t-1}} & \frac{\partial z_{2,t}}{\partial z_{2,t-1}} & 0 & \frac{\partial z_{2,t}}{\partial \epsilon_{2,t}} \end{bmatrix}.$$

Given that this Jacobian is triangular, we can efficiently compute its determinant as $\prod_i \frac{\partial z_{i,t}}{\partial \epsilon_{i,t}}$. Furthermore, because the noise terms are mutually independent, and hence $\epsilon_{i,t} \perp \epsilon_{j,t}$ for $j \neq i$ and $\epsilon_t \perp \mathbf{z}_{t-1}$, we can write the RHS of Eq. 18 as:

$$\begin{aligned} p(z_{1,t-1}, z_{2,t-1}, z_{1,t}, z_{2,t}) &= p(z_{1,t-1}, z_{2,t-1}) \times p(\epsilon_{1,t}, \epsilon_{2,t}) / \left| \det \mathbf{J_f} \right| \quad \text{(because } \epsilon_t \perp \mathbf{z}_{t-1}) \\ &= p(z_{1,t-1}, z_{2,t-1}) \times \prod_i p(\epsilon_{i,t}) / \left| \det \mathbf{J_f} \right| \quad \text{(because } \epsilon_{1,t} \perp \epsilon_{2,t}) \end{aligned}$$

(19)

Finally, by canceling out the marginals of the lagged latent variables $p(z_{1,t-1}, z_{2,t-1})$ on both sides, we can evaluate the transition prior likelihood as:

$$p(z_{1,t}, z_{2,t} | z_{1,t-1}, z_{2,t-1}) = \prod_i p(\epsilon_{i,t}) / \left| \det \mathbf{J_f} \right| = \prod_i p(\epsilon_{i,t}) \times \left| \det \mathbf{J_f}^{-1} \right|. \tag{20}$$

Now we generalize this example and derive the prior likelihood below.

Let $\{f_i^{-1}\}_{i=1,2,3\ldots}$ be a set of learned inverse transition functions that take the estimated latent causal variables, and output the noise terms, i.e., $\hat{\epsilon}_{i,t} = f_i^{-1}\left(\hat{z}_{i,t}, \{\hat{\mathbf{z}}_{t-\tau}, c_t\}\right)$.

Design transformation $\mathbf{A} \rightarrow \mathbf{B}$ with low-triangular Jacobian as follows:

$$\underbrace{\left[\hat{\mathbf{z}}_{t-L}, \ldots, \hat{\mathbf{z}}_{t-1}, \hat{\mathbf{z}}_t\right]^\top}_{\mathbf{A}} \text{ mapped to } \underbrace{\left[\hat{\mathbf{z}}_{t-L}, \ldots, \hat{\mathbf{z}}_{t-1}, \hat{\epsilon}_{i,t}\right]^\top}_{\mathbf{B}}, \textit{ with } \mathbf{J_{A \rightarrow B}} = \begin{pmatrix} \mathbb{I}_{nL} & 0 \\ * & \text{diag}\left(\frac{\partial f_{i,j}^{-1}}{\partial \hat{z}_{jt}}\right) \end{pmatrix}.$$

(21)

Similar to Eq. 20, we can obtain the joint distribution of the estimated dynamics subspace as:

$$\log p(\mathbf{A}) = \underbrace{\log p\left(\hat{\mathbf{z}}_{t-L}, \ldots, \hat{\mathbf{z}}_{t-1}\right) + \sum_{j=1}^{n} \log p(\hat{\epsilon}_{i,t})}_{\text{Because of mutually independent noise assumption}} + \log\left(\left|\det\left(\mathbf{J_{A \rightarrow B}}\right)\right|\right). \tag{22}$$

$$\log p\left(\hat{\mathbf{z}}_t | \{\hat{\mathbf{z}}_{t-\tau}\}_{\tau=1}^{L}, c_t\right) = \sum_{j=1}^{n} \log p(\hat{\epsilon}_{i,t} | c_t) + \sum_{i=j}^{n} \log\left|\frac{\partial f_i^{-1}}{\partial \hat{z}_{i,t}}\right| \tag{23}$$

## B.3 Derivation of ELBO

Then the second part is to maximize the Evidence Lower BOund (ELBO) for the VAE framework, which can be written as:

$$
\begin{aligned}
\text{ELBO} &\triangleq \log p_{\text{data}}(\mathbf{X}) - D_{KL}(q_\phi(\mathbf{Z}|\mathbf{X})||p_{\text{data}}(\mathbf{Z}|\mathbf{X})) \\
&= \mathbb{E}_{\mathbf{Z}\sim q_\phi(\mathbf{Z}|\mathbf{X})} \log p_{\text{data}}(\mathbf{X}|\mathbf{Z}) - D_{KL}(q_\phi(\mathbf{Z}|\mathbf{X})||p_{\text{data}}(\mathbf{Z}|\mathbf{X})) \\
&= \mathbb{E}_{\mathbf{Z}\sim q_\phi(\mathbf{Z}|\mathbf{X})} \log p_{\text{data}}(\mathbf{X}|\mathbf{Z}) - \mathbb{E}_{\mathbf{Z}\sim q_\phi(\mathbf{Z}|\mathbf{X})}\left[\log q_\phi(\mathbf{Z}|\mathbf{X}) - \log p_{\text{data}}(\mathbf{Z})\right] \\
&= \mathbb{E}_{\mathbf{Z}\sim q_\phi(\mathbf{Z}|\mathbf{X})}\left[\log p_{\text{data}}(\mathbf{X}|\mathbf{Z}) + \underbrace{\log p_{\text{data}}(\mathbf{Z})}_{\mathbb{E}_{\mathbf{c}}\left[\sum_{t=1}^{T}\log p(\mathbf{z}_t|\mathbf{z}_{t-1},c_t)\right]} - \log q_\phi(\mathbf{Z}|\mathbf{X})\right] \\
&= \mathbb{E}_{\mathbf{z}_t}\left[\underbrace{\sum_{t=1}^{T}\log p_{\text{data}}(\mathbf{x}_t|\mathbf{z}_t)}_{-\mathcal{L}_{\text{Recon}}} + \underbrace{\mathbb{E}_{\mathbf{c}}\left[\sum_{t=1}^{T}\log p_{\text{data}}(\mathbf{z}_t|\mathbf{z}_{\text{Hx}},c_t)\right] - \sum_{t=1}^{T}\log q_\phi(\mathbf{z}_t|\mathbf{x}_t)}_{-\mathcal{L}_{\text{KLD}}}\right]
\end{aligned}
\tag{24}
$$

## B.4 Synthetic Dataset Generation

We generated two synthetic datasets (A and B) with different nonlinear mixing functions. In this section we will introduce the detailed implementation of the generation. The generation can be split into steps (1) sample $c_t$ from a Markov chain, (2) generate $\mathbf{z}_t$ with different transition functions $f_{c_t}$ with respect to $c_t$, and (3) generate observation $\mathbf{x}_t$ via mixing function $\mathbf{g}$.

### B.4.1 Sample $c_t$ from Markov chain

We first randomly initialized a Markov chain with transition matrix $\mathbf{A}$ and sample 20,000 steps.

### B.4.2 Generation of latent variables $\mathbf{z}_t$

We first randomly initialized $|C| = 5$ different transition functions $\{f_1, f_2, \ldots, f_{|C|}\}$ with different MLPs, and generate $\mathbf{z}_t = f_{c_t}(\mathbf{z}_{\text{Hx}})$. The dimensions are set to 8 for fair comparison.

### B.4.3 Generation of observations $\mathbf{x}_t$

The difference between datasets A and B is the mixing function. We use a two-layer randomly initialized MLP for dataset A and a three-layer MLP for dataset B. For each linear layer in the MLP, we use condition number of the weight matrix to filter out ones that are not "invertible".

## B.5 Modified CartPole Dataset Generation

Similar to the synthetic datasets, we also sample from a Markov chain and get $c_t$. For the modified CartPole, we initialized 5 different environments which have different combinations of hyperparameters such as gravity, pole mass, etc. A detailed comparison is listed in Table 1.

Table 1: Different configs for different Modified CartPole environments.

| Environment ID | Gravity | Pole Mass | Noise Scale |
|:---:|:---:|:---:|:---:|
| 0 | 9.8 | 0.2 | 0.01 |
| 1 | 24.79 | 0.5 | 0.01 |
| 2 | 3.7 | 1.0 | 0.01 |
| 3 | 11.15 | 1.5 | 0.01 |
| 4 | 0.62 | 2.0 | 0.01 |

At each time step $t$ the environment will load the corresponding hyperparameters for given $c_t$ and update the states $\mathbf{z}_t$ according to the configuration given $c_t$. The nonlinear mixing function from states to observations $\mathbf{x}_t$ is fixed by a rendering method in the gym package.

## B.6 MoSeq Dataset

In the MoSeq dataset, the observations $\mathbf{x}_t$ are taken to be the first 10 principal components of depth camera video data of mice exploring an open field. The dataset consists of 20-minute depth camera recordings of 24 mice. In preprocessing, the videos are cropped and centered around the mouse centroid and then filtered to remove recording artifacts. Finally, the preprocessed video is projected onto the top principal components to obtain a 10-dimensional time series.

## B.7 Mean Correlation Coefficient

MCC is a standard metric for evaluating the recovery of latent factors in ICA literature. MCC first calculates the absolute values of the correlation coefficient between every ground-truth factor against every estimated latent variable. Pearson correlation coefficients or Spearman's rank correlation coefficients can be used depending on whether componentwise invertible nonlinearities exist in the recovered factors. The possible permutation is adjusted by solving a linear sum assignment problem in polynomial time on the computed correlation matrix.

## B.8 Network Architecture

We summarize our network architecture below and describe it in detail in Table 2 and Table 3.

Table 2: Architecture details. BS: batch size, T: length of time series, i_dim: input dimension, z_dim: latent dimension, LeakyReLU: Leaky Rectified Linear Unit.

| Configuration | Description | Output |
|---|---|---|
| **ARHMM** | Autoregressive HMM for Synthetic Data | |
| Input: $\mathbf{x}_{1:T}$ | Observed time series | BS $\times$ T $\times$ i_dim |
| Emission Module | Compute $\mu_{\mathbf{z}_{t+1}}, \sigma_{\mathbf{z}_{t+1}}$ | BS $\times$ T $\times$ 2 $\times$ z_dim |
| **MLP-Encoder** | Encoder for Synthetic Data | |
| Input: $\mathbf{x}_{1:T}$ | Observed time series | BS $\times$ T $\times$ i_dim |
| Dense | 128 neurons, LeakyReLU | BS $\times$ T $\times$ 128 |
| Dense | 128 neurons, LeakyReLU | BS $\times$ T $\times$ 128 |
| Dense | 128 neurons, LeakyReLU | BS $\times$ T $\times$ 128 |
| Dense | Temporal embeddings | BS $\times$ T $\times$ z_dim |
| **MLP-Decoder** | Decoder for Synthetic Data | |
| Input: $\hat{\mathbf{z}}_{1:T}$ | Sampled latent variables | BS $\times$ T $\times$ z_dim |
| Dense | 128 neurons, LeakyReLU | BS $\times$ T $\times$ 128 |
| Dense | 128 neurons, LeakyReLU | BS $\times$ T $\times$ 128 |
| Dense | i_dim neurons, reconstructed $\hat{\mathbf{x}}_{1:T}$ | BS $\times$ T $\times$ i_dim |
| **Factorized Inference Network** | Bidirectional Inference Network | |
| Input | Sequential embeddings | BS $\times$ T $\times$ z_dim |
| Bottleneck | Compute mean and variance of posterior | $\mu_{1:T}, \sigma_{1:T}$ |
| Reparameterization | Sequential sampling | $\hat{\mathbf{z}}_{1:T}$ |
| **Prior Network** | Nonlinear Transition Prior Network | |
| Input | Sampled latent variable sequence $\hat{\mathbf{z}}_{1:T}$ | BS $\times$ T $\times$ z_dim |
| InverseTransition | Compute estimated residuals $\hat{\epsilon}_{it}$ | BS $\times$ T $\times$ z_dim |
| JacobianCompute | Compute $\log\left(\left\lvert\det\left(\mathbf{J}\right)\right\rvert\right)$ | BS |

Table 3: Architecture details on CNN encoder and decoder. BS: batch size, T: length of time series, h_dim: hidden dimension, z_dim: latent dimension, F: number of filters, (Leaky)ReLU: (Leaky) Rectified Linear Unit.

| Configuration | Description | Output |
|---|---|---|
| **CNN-Encoder** | Feature Extractor | |
| Input: $\mathbf{x}_{1:T}$ | RGB video frames | BS $\times$ T $\times$ 3 $\times$ 64 $\times$ 64 |
| Conv2D | F: 32, BatchNorm2D, LeakyReLU | BS $\times$ T $\times$ 32 $\times$ 64 $\times$ 64 |
| Conv2D | F: 32, BatchNorm2D, LeakyReLU | BS $\times$ T $\times$ 32 $\times$ 32 $\times$ 32 |
| Conv2D | F: 32, BatchNorm2D, LeakyReLU | BS $\times$ T $\times$ 32 $\times$ 16 $\times$ 16 |
| Conv2D | F: 64, BatchNorm2D, LeakyReLU | BS $\times$ T $\times$ 64 $\times$ 8 $\times$ 8 |
| Conv2D | F: 64, BatchNorm2D, LeakyReLU | BS $\times$ T $\times$ 64 $\times$ 4 $\times$ 4 |
| Conv2D | F: 128, BatchNorm2D, LeakyReLU | BS $\times$ T $\times$ 128 $\times$ 1 $\times$ 1 |
| Dense | F: 2 * z_dim = dimension of hidden embedding | BS $\times$ T $\times$ 2 * z_dim |
| **CNN-Decoder** | Video Reconstruction | |
| Input: $\mathbf{z}_{1:T}$ | Sampled latent variable sequence | BS $\times$ T $\times$ z_dim |
| Dense | F: 128 , LeakyReLU | BS $\times$ T $\times$ 128 $\times$ 1 $\times$ 1 |
| ConvTranspose2D | F: 64, BatchNorm2D, LeakyReLU | BS $\times$ T $\times$ 64 $\times$ 4 $\times$ 4 |
| ConvTranspose2D | F: 64, BatchNorm2D, LeakyReLU | BS $\times$ T $\times$ 64 $\times$ 8 $\times$ 8 |
| ConvTranspose2D | F: 32, BatchNorm2D, LeakyReLU | BS $\times$ T $\times$ 32 $\times$ 16 $\times$ 16 |
| ConvTranspose2D | F: 32, BatchNorm2D, LeakyReLU | BS $\times$ T $\times$ 32 $\times$ 32 $\times$ 32 |
| ConvTranspose2D | F: 32, BatchNorm2D, LeakyReLU | BS $\times$ T $\times$ 32 $\times$ 64 $\times$ 64 |
| ConvTranspose2D | F: 3, estimated scene $\hat{\mathbf{x}}_{1:T}$ | BS $\times$ T $\times$ 3 $\times$ 64 $\times$ 64 |

