# OpenReview forum: "Temporally Disentangled Representation Learning under Unknown Nonstationarity"
_NeurIPS.cc/2023/Conference — NeurIPS 2023 poster_

### Official Review · Reviewer_PKrt · 2023-07-02

**Soundness:** 3 good
**Presentation:** 4 excellent
**Contribution:** 3 good
**Rating:** 7
**Confidence:** 3

**Summary:**

This paper addresses the problem of identifying latents representations from sequential data which is stationary within contexts, and non-stationary between contexts. Existing work has either conditioned on observed auxiliary variables that indicate which context one is in (e.g. time contrastive learning and the followups), or they relied on mutually (conditionally) independent latents and did not allow time-delayed latent causal influences. The paper presents an approach called Nonstationary Temporal Disentangled Causal Representation Learning (NTDC)and shows conditions under which they can identify time-delayed latent causal variables and their relations without the need for observed auxiliary variables. They formulate the problem as a discrete Markov process and establish identifiability of the latent independent components. They give impressive experimental results on both synthetic and real-world datasets to demonstrate the proposed method's success in recovering latent variables.

**Strengths:**

* Really nice experimental results section. The results on simulated data and I appreciated the section on the MoSeq dataset analyzing mouse behaviour.
* I think the graphical model that the paper analyses is very practically useful for abstracting time series data. By breaking a time series into conditionally stationary domains, $c_t$, that change more slowly, you gain a more interpretable model of the underlying behaviour of the system (the mouse experiments are nice illustration of this). Empirically, we've seen results similar to this before in the original MoSeq paper, but it is nice to have identifiability results to go with these approaches.

**Weaknesses:**

* Assumptions (6) and (7) in Theorem 2 are unintuitive to the point that I'm not sure the theorem is useful at all. The main value of identifiability theorems are to give a set of preconditions on the data under which a practitioner can expect a disentanglement routine would work. But I'm not sure that there are any practitioners who would be able to judge whether those assumptions hold in their data, and there was no attempt to give any intuition for when they might hold and when they would fail.

**Questions:**

1. Can you give any more intuitive assumptions that are sufficient for Theorem 2 to hold? Or give examples of distributions for which they hold and examples for which they fail? If done well, this would make the theory section stronger.

**Limitations:**

The paper already does a reasonable job of outlining the limitations of their work.

---

> ### Author Rebuttal · Authors · 2023-08-10
>
> We are sincerely grateful to the reviewer for the informative feedback. Please see our point-to-point response below.
>
> > Weaknesses: ... But I'm not sure that there are any practitioners who would be able to judge whether those assumptions hold in their data ...
>
> Firstly we would like to mention that our assumed conditions are generally testable. One can run experiments with multiple seeds and compute the similarity (MCCs) between the learned representations; if the representation is not unique, then our assumptions must be violated.
>
> > Q1: Can you give any more intuitive assumptions that are sufficient for Theorem 2 to hold? Or give examples of distributions for which they hold and examples for which they fail? If done well, this would make the theory section stronger.
>
> This assumption was first introduced in GCL[1], namely, "sufficient variability", to extend the modulated exponential families [2] to general modulated distributions. Essentially, the condition says that the nonstationary regimes $c$ must have a sufficiently complex and diverse effect on the transition distributions. In other words, if the underlying distributions are composed of relatively many domains of data, the condition generally holds true. For instance, in the linear Auto-Regressive (AR) model with Gaussian innovations where only the noise variance changes, the condition reduces to the statement in [3] that the variance of each noise term fluctuates somewhat independently of each other in different nonstationary regimes. Then the condition is easily attained if the variance vector of noise terms in any regime is not a linear combination of variance vectors of noise terms in other regimes.
>
> We further illustrate the condition using the example of modulated conditional exponential families in [1]. Let the log-pdf $q(\mathbf{z}_t \vert \mathbf{z}\_{\text{Hx}}, c)$ be a conditional exponential family distribution of order $k$ given nonstationary regime $c$ and history $\mathbf{z}\_{\text{Hx}}$:
>
> $$q(z_{it} \vert \mathbf{z}\_{\text{Hx}}, c) = q_i(z_{it}) + \sum_{j=1}^k q_{ij}(z_{it}) \lambda_{ij}(\mathbf{z}\_{\text{Hx}}, c) - \log Z(\mathbf{z}\_{\text{Hx}}, c),$$
>
> where $q_i$ is the base measure, $q_{ij}$ is the function of the sufficient statistic, $\lambda_{ij}$ is the natural parameter, and $\log Z$ is the log-partition. Loosely speaking, the sufficient variability holds if the modulation of by $c$ on the conditional  distribution $q(z_{it} \vert \mathbf{z}_{\text{Hx}}, c)$ is not too simple in the following sense:
>
> 1. Higher order of $k$ ($k>1$) is required. If $k=1$, the sufficient variability cannot hold;
> 2. The modulation impacts $\lambda_{ij}$ by $c$ must be linearly independent across regimes $c$. The sufficient statistics functions $q_{ij}$ cannot be all linear, i.e., we require higher-order statistics.
>
> Further details of this example can be found in Appendix B of [1]. In summary, we need the modulation by $c$ to have diverse (i.e., distinct influences) and complex impacts on the underlying data generation process.
>
> # References:
>
> [1] Hyvarinen, Aapo, Hiroaki Sasaki, and Richard Turner. "Nonlinear ICA using auxiliary variables and generalized contrastive learning." The 22nd International Conference on Artificial Intelligence and Statistics. PMLR, 2019.
>
> [2] Hyvarinen, Aapo, and Hiroshi Morioka. "Unsupervised feature extraction by time-contrastive learning and nonlinear ica." Advances in neural information processing systems 29 (2016).
>
> [3] Matsuoka, Kiyotoshi, Masahiro Ohoya, and Mitsuru Kawamoto. "A neural net for blind separation of nonstationary signals." Neural networks 8.3 (1995): 411-419.

---

> > ### Author Response · Authors · 2023-08-13
> >
> > Thank you again for your insightful comments on our paper. If you think our response needs further clarification, or if you have additional questions, please don't hesitate to let us know. Your feedback is highly valued, and we're ready to engage in further discussion if needed.

---

> > ### Comment · Reviewer_PKrt · 2023-08-17
> > **Sufficient variability**
> >
> > Thanks for the clarification. This discussion is useful and should be in the paper: I'm someone who knows this literature fairly well, and while it looked like a sufficient variability-style assumption, I missed that it was exactly sufficient variability. Despite the fact that sufficient variability is now a commonly used assumption, I think a description of what it requires and when it fails should still be in every paper that uses it so that the paper has a self-contained explanation of when we can expect the method to work / fail.
> >
> > On the assumption that you'll add this discussion to the paper, and I've increased my score to a 7.

---

> ### Author Response · Authors · 2023-08-17
> **Thanks for your feedback and updating the score to 7**
>
> Dear Reviewer PKrt,
>
> Thanks for your recognition of our rebuttal efforts and kindly updating the score. We are very delighted to hear that all of your significant concerns have been resolved.
>
> With best regards,
>
> Authors of submission 9100

---

### Official Review · Reviewer_eHJB · 2023-07-09

**Soundness:** 4 excellent
**Presentation:** 3 good
**Contribution:** 3 good
**Rating:** 6
**Confidence:** 3

**Summary:**

This paper gives identifiability results in a new setting, along with an estimation method and experiments validating the estimation method.

There are existing identifiability results in the nonstationary setting if you assume that the auxiliary variables are observed. There are other existing identifiability results in the nonstationary setting if you assume that the auxiliary variables evolve as a Markov chain but the latents only affect the current timestep. This paper gives an identifiability result in the nonstationary setting where the auxiliary variables are *not* observed and the auxiliary variables evolve as a Markov chain but the latent variables can affect the observed variables across time steps.

The estimation method is an extension of Sequential VAEs, where there is an autoregressive Hidden Markov Model model that models the nonstationarity, a prior network that estimates the prior using a conditional normalizing flow, and a VAE where the encoder fits the demixing function and the decoder fits the mixing function.

They perform experiments on two synthetic datasets that satisfy their identifiability conditions, a video dataset for a modified Cartpole environment, and a video dataset of mouse behavior. For the synthetic datasets, they compare MCC with seven baselines (BetaVAE, iVAE, TVL, SlowVAE, PCL, TDRL, and HMNLICA). For the video cartpole dataset, they compare MCC against one baseline (TDRL). For the mouse video dataset they don’t have ground-truth independent components so instead of numerical results they provide a visualization of the recovered components.

**Strengths:**

Strengths:
- I am not especially up-to-date on identifiability literature, but as far as I am aware this is a novel problem setting and novel result.
- I agree that the “observed auxiliary variable” condition in many nonlinear ICA identifiability results is not realistic, and it is a useful direction to prove results that do not rely on this assumption.
- The choice of baselines is very thorough for the synthetic datasets.
- They use two video datasets to demonstrate their estimation method in a more realistic setting than synthetic data, which I appreciate since this is mostly a theory paper.

**Weaknesses:**

Weaknesses:
- The writing is okay but not great. It gets the point across but it could use proofreading and better word choices in some places. Ex: “It is obvious that the causal relations among those latent variables are more meaningful and the identifiability is urgently needed.” <- this sentence could be improved.
- See questions below.

**Questions:**

- In the Cartpole experiments, why do you only provide Accuracy and MSE A metrics for your method and not for baseline methods? And can you provide evaluations for the other baselines for this experiment?


**Limitations:**

- As they mention in the conclusion section, they assume that the nonstationary variables follow a Markov chain, which is a somewhat restrictive assumption.
- No concerns about negative societal impacts.

---

> ### Author Rebuttal · Authors · 2023-08-10
>
> We appreciate that you read our paper very carefully and your informative feedback, which has helped improve our paper. Below please see our response to your concerns.
>
> >Q1: In the Cartpole experiments, why do you only provide Accuracy and MSE A metrics for your method and not for baseline methods? And can you provide evaluations for the other baselines for this experiment?
>
> A1: Baselines either ignore the nonstationarity in the system (completely ignore $c_t$) or require domain indices ($c_t$) to be observed. In both scenarios, baselines don't predict $c_t$ as their output hence it doesn't make sense to evaluate the accuracy for $c_t$ and the transition matrix $\mathbf{A}$. It worth to mention that for HMNLICA, even though it estimates $c_t$, it doesn't consider the time-delayed relation for latent variables $\mathbf{z}_t$, for real datasets such as CartPole, the algorithm doesn't converge during training, hence we dropped it in our experiments.
>
> >The writing is okay but not great. It gets the point across but it could use proofreading and better word choices in some places. Ex: “It is obvious that the causal relations among those latent variables are more meaningful and the identifiability is urgently needed.” <- this sentence could be improved.
>
> We thanks the reviewer for pointing this out, we update the sentence as follows: ''Learning causal relations has practical use case, which benefits a lot of downstream tasks''. We will also proofread the paper and improve the writing in the final version. If you have any further suggestions please let us know in the discussion period.

---

> > ### Author Response · Authors · 2023-08-13
> >
> > Thank you again for your valuable comments on our paper. Should you require more information or have further concerns, please do not hesitate to let us know. We are more than willing to engage in further discussions, provide additional information, or clarify our methodologies. Your insights are instrumental in improving our work, and we sincerely value your expertise and time. We look forward to any further comments you may have.

---

> > ### Comment · Area_Chair_BrYY · 2023-08-18
> > **Thanks**
> >
> > Thank you for this feedback authors. This will be taken into account.

---

> > > ### Comment · Reviewer_eHJB · 2023-08-19
> > > **Reviewer response to rebuttal**
> > >
> > > Thanks to the authors for your responses to my questions and comments about the experiments and baseline comparisons and about the writing. I don't have any further questions and I'll keep my score as is.

---

> > > > ### Author Response · Authors · 2023-08-19
> > > > **Thanks for recommending acceptance to our paper**
> > > >
> > > > Dear Reviewer eHJB,
> > > >
> > > > Thanks for your recognition of our rebuttal efforts and keep recommending acceptance to our paper. We are very delighted to hear that all of your concerns and questions have been addressed.
> > > >
> > > > With best regards,
> > > >
> > > > Authors of submission 9100

---

### Official Review · Reviewer_5DXD · 2023-07-21

**Soundness:** 2 fair
**Presentation:** 1 poor
**Contribution:** 2 fair
**Rating:** 3
**Confidence:** 3

**Summary:**

This paper presents identifiability results that handle nostationary time-delayed causally-related processes without auxiliary variables. In addition, the authors propose NTDC, which is a neural network that is based on their identifiability results. The method is evaluated on a few tasks in comparison to a few baselines, showing significance of toy tasks generated by the authors.


**Strengths:**

This paper introduces identifiability results on sequential information which is causally-related without additional variables. Thus, this paper extends prior work from a theoretic viewpoint. Another strength of the paper is the new deep framework that incorporates the theoretical results.


**Weaknesses:**

One of the main weaknesses of the paper is its presentation. In particular, a lot of jargon is used without properly introducing the terms. For instance, domain indices are used throughout the paper without ever defining what it means. Similarly, Sec. 4.1 is missing a lot of details (what is ARHMM? what is the prior network?). Sec. 4.2 seems to be unfinished in terms of writing. Missing details and descriptions make it hard to properly evaluate the proposed NTDC and position it with respect to prior work.

Another main shortcoming of the work is the lack of discussion on related work for sequential disentanglement. Only paper [20] is mentioned (extremely) briefly. However, there are already at least ten papers or more since [20] which significantly improved the state-of-the-art results, the neural network models, the evaluation and theory.

Another shortcoming of the paper is the dicsussion in the introduction. In particular, the authors argue their mouse example might be challenging for existing work. In addition, they argue to be first to consider scenario Fig. 1c. However, in a recent ICLR'23 paper with the title "Multifactor Sequential Disentanglement via Structured Koopman Autoencoders" (SKD) by Berman et al. the authors show a framework that considers a similar model to Fig. 1c, and it is able to handle data such as the mouse example. The authors should compare their work with SKD both qualitatively and quantitatively. Similarly, the work "Contrastively Disentangled Sequential Variational Audoencoder" (C-DSVAE) by Bai et al. can also deal with a similar scenario and should be compared with.

Another shortcoming of the paper is its evaluation. In particular, the proposed method is compared with several baselines only on toy datasets that the authors create. No sequential disentanglement baselines are considered. On real world data, the authors only compare with a single method.


**Questions:**

See above.

**Limitations:**

See above.

---

> ### Author Rebuttal · Authors · 2023-08-10
>
> We thank the reviewer for the insightful review and valuable feedback. We respond to your concerns point-by-point below.
>
> > Q1.1: One of the main weaknesses of the paper is its presentation. In particular, a lot of jargon is used without properly introducing the terms. For instance, domain indices are used throughout the paper without ever defining what it means.
>
> Domain indices is the discrete variable that denote different domains. In our model it is $c_t$ which is introduced in Sec 2.1 Eq (2).
>
> We will add more explanation to the technical terms and if there is any further unclear issues, please kindly let us know. We are more than happy to provide more details during the discussion.
>
> > Q1.2: Similarly, Sec. 4.1 is missing a lot of details (what is ARHMM? what is the prior network?).
>
> We thank the reviewer for raising this question. ARHMM refers to Autoregressive Hidden Markov Module (which is mentioned in line 193-194), which is a standard abbreviation. Such module is to model the nonstationarity with input of observed data $\mathbf{x}_t$ and output the estimated domain indices $c_t$. Prior Network (introduced in line 202-204) refers to the module to learn transition priors, such priors are obtained by first learning inverse transition functions $f_z^{-1}$ that take the estimated latent variables and output random noise terms, and applying the change of variables formula to the transformation
>
> $p(\hat{z}\_{t}\vert \hat{\mathbf{z}}\_{\text{Hx}}, c_t) = p_{\epsilon_c}\left(\hat{f}\_{z}^{-1}(\hat{z}\_{t}, \hat{\mathbf{z}}\_{\text{Hx}}, \hat{\boldsymbol{\theta}}\_{c_t})\right)\Big|\frac{\partial \hat{f}_z^{-1}}{\partial \hat{z}\_{t}}\Big|$.
>
> And the Encoder-Decoder is a standard module in Variational Auto-Encoder based method which projects the $\mathbf{x}_t$ to latent space $\mathbf{z}_t$ while preserving the ability to reconstruct $\mathbf{x}_t$.
>
> Please let us know if the information above make it easier for you to understand, we are also happy to add more detailed explanations if you have further questions.
>
> > Q1.3: Sec. 4.2 seems to be unfinished in terms of writing. Missing details and descriptions make it hard to properly evaluate the proposed NTDC and position it with respect to prior work.
>
> We apologize for accidentally commenting out part of the last sentence in Sec 4.2 in the submitted version. The complete writing is updated as follows: ''Specifically, we obtain the log-likelihood of the posterior, evaluate the prior
>
> $\log p\left(\hat{\mathbf{z}}_t \vert \hat{\mathbf{z}}\_{\text{Hx}}, c_t\right)$ in Eq. (9), and compute their mean difference in the dataset as the KL loss:
>
> $\mathcal{L}\_{\text{KLD}} = \mathbb{E}_{\mathbf{\hat z}_t \sim q\left(\mathbf{\hat z}_t \vert \mathbf{x}_t\right)} \log q(\mathbf{\hat z}_t|\mathbf{x}_t) - \log p\left(\hat{\mathbf{z}}_t \vert \hat{\mathbf{z}}\_{\text{Hx}}, c_t\right)$.''
>
> > Q2, Q3: ... lack of discussion on related work for sequential disentanglement ... However, in a recent ICLR'23 paper with the title "Multifactor Sequential Disentanglement via Structured Koopman Autoencoders" (SKD) by Berman et al. the authors show a framework that considers a similar model to Fig. 1c, and it is able to handle data such as the mouse example. The authors should compare their work with SKD both qualitatively and quantitatively. Similarly, the work "Contrastively Disentangled Sequential Variational Audoencoder" (C-DSVAE) by Bai et al. can also deal with a similar scenario and should be compared with.
>
> We thank the reviewer for pointing out and we thank the reviewer for providing related work with references. We will definitely add the related topics in related work section. As for the relation between ''Sequential Disentanglement'' and our setting, we kindly remind that we would like to investigate the disentanglement from the causal lens which means we care about recovering the ground truth of the data generating process. In other words, there are multiple ways to disentangle the variable and obtain well-disentangled representations, but only a very small subset of them are actually following the ground truth i.e. achieving identifiability. The main contribution for this work is slightly different from just finding a disentangled representation under nonstationary setting. Instead we care more about if we can recover the ground truth and provided the identifiability, hence the Encoder-Decoder module can adopt any current state-of-the-art method such as SKD, and such choice is orthogonal to our contribution.
>
> Despite of the difference mentioned above, we still compared SKD in the CartPole dataset and reported the MCC in the table below. Note, and also as mentioned in SKD paper that ''C-DSVAE'' uses slightly more supervision in data augmentation which requires more knowledge than purely unsupervised learning, for fair comparison we only conducted our additional experiment using SKD. As shown in table below, the MCC for SKD is better than a variety of baselines, however, we can see the distinction between well-disentangled models and identifiable models, only the models with identifiability can find the ground truth latent variables with theoretical guarantee.
>
> | Method | MCC |
> | :-: | - |
> | BetaVAE | 57.54 |
> | i-VAE | 60.14 |
> | TCL | 65.07 |
> | SlowVAE | 63.16 |
> | SKD | 73.24 |
> |NTDC|96.06|
>
> > Q4: Another shortcoming of the paper is its evaluation. In particular, the proposed method is compared with several baselines only on toy datasets that the authors create. No sequential disentanglement baselines are considered. On real world data, the authors only compare with a single method.
>
> We additionally compared baseline methods together with SKD in the CartPole dataset and showed the result in previous question.
>
> Please let us know if you have further concerns and we are happy to provide more detailed information in the discussion phase.

---

> > ### Comment · Reviewer_5DXD · 2023-08-11
> >
> > Thank you.

---

> > > ### Author Response · Authors · 2023-08-13
> > >
> > > Thank you once again for your insightful comments and invaluable advice, which helped us improve the paper's quality and clarity. We are committed to demonstrating the merits of our paper and would be pleased to engage further with you. Should there be a need for any additional discussion or clarification that may enhance the paper's value, please don't hesitate to let us know. We appreciate your comments and advice.

---

> > > ### Author Response · Authors · 2023-08-17
> > > **Would you like to consider updating your recommendation?**
> > >
> > > Dear Reviewer 5DXD
> > >
> > > Thank you for your prompt feedback. Would you like to consider updating your recommendation, if your concerns are properly addressed?
> > >
> > > With best regards,
> > >
> > > Authors of submission 9100

---

> > > > ### Author Response · Authors · 2023-08-21
> > > >
> > > > Dear Reviewer 5DXD,
> > > >
> > > > Once again, we are grateful for your time.  We have been eagerly waiting for your feedback on our point-to-point response.  Since the discussion period will end in shortly, we will be online waiting to see whether your previous concern was properly addressed. We understand you have a busy schedule, but would highly appreciate it if you could take into account our response when updating the rating and having discussions with AC and other reviewers.
> > > >
> > > > Thanks a lot for your contribution to NeurIPS 2023,
> > > >
> > > > Authors of #9100

---

### Official Review · Reviewer_t3u7 · 2023-07-30

**Soundness:** 2 fair
**Presentation:** 2 fair
**Contribution:** 2 fair
**Rating:** 4
**Confidence:** 3

**Summary:**

The study focuses on unsupervised representation learning for sequential data with time-delayed causal influences. Identifiability results for disentangling causally-related latent variables have been established in stationary settings using temporal structure. However, existing work only partially addresses this in nonstationary settings by using observed auxiliary variables or introducing the Markov assumption, but not both simultaneously. The authors introduce NTDC, to reconstruct time-delayed latent causal variables and identify their relations without the need for auxiliary variables, by utilizing deep generative models. The experimental results demonstrate that the proposed methodology outperforms existing baselines by effectively exploiting nonstationarity and distinguishing distribution shifts.

**Strengths:**

- The paper is clearly written.
- The authors tackle the setting of previous works and propose a non-existing work, which is described in Figure 1.
- The theoretical grounds are sufficiently provided with proper theorems.

**Weaknesses:**

- There are some grammatical errors in the main paper.
- There is no explanation of the "disentangled" which is included both in the proposed model name and the title of the manuscript.
- The authors provide their code with anonymity, but there is limited explanations of the codes that which is directly linked to the proposed NTDC and how one can run the code.

**Questions:**


- Could you provide more motivational examples of the model structure in Figure 1(d) other than the mouse movement example?
- Are there any derivation with probabilistic graphical model settings of Figure 1(d)? I wonder whether the problem could be solved in a statistical manner rather than just utilizing DGMs.
- I guess the model structure can be viewed as a doubly-hidden Markov model, or it could be more generalized as a multi-layer hidden Markov model. Could proposed theorems, as well as the proposed NTDC, be generalized in such settings?
- What's the specific meaning of "disentangled" representation in the title as well as the model name? And how the disentangled representation is achieved?


**Limitations:**

- There are no potential negative societal impacts of their work.
- Even though the authors provided a new work, my concern is that it lacks significance considering the NeurIPS standard.

---

> ### Author Rebuttal · Authors · 2023-08-10
>
> We thank the reviewer for the insightful review and valuable feedback. We respond to your concerns point-by-point below.
> > Q1: Could you provide more motivational examples of the model structure in Figure 1(d) other than the mouse movement example?
>
> Here are some examples with explanation:
> 1. For general time series data such as stock market data, the underlying dynamics (map $\mathbf{z}\_{t-1} \rightarrow \mathbf{z}\_{t}$) changes dramatically across time. Such a nonstationary process can be modeled by Figure 1(d) by choosing different $c_t$.
> 2. Videos have similar properties. $c_t$ represents the events in the video. Within the same event, the changing dynamics stay the same, and across different events, the changing dynamics are different.
>
> > Q2: Are there any derivation with probabilistic graphical model settings of Figure 1(d)? I wonder whether the problem could be solved in a statistical manner rather than just utilizing DGMs.
>
> Thanks for the question, we are a little bit confused by the terminology here. Do you want to say PGMs instead of DGMs? From our understanding, using PGM doesn't conflicts with the solving the problem in a statistical manner. It is just to show the underlying dependence relations among the variables, and finally the problem is also solved in the statistical manner with the help of PGMs. Please let us know if there is any misinterpretation for this question, we are very happy to provide more explanations during the discussion.
>
> > Q3: I guess the model structure can be viewed as a doubly-hidden Markov model, or it could be more generalized as a multi-layer hidden Markov model. Could proposed theorems, as well as the proposed NTDC, be generalized in such settings?
>
> Thanks for your good observation and question. Instead of a doubly-hidden Markov model, our model is more similar to a nonstationary state space model with a HMM component on top of the hidden variables $\mathbf{z}\_t$. As for the generalization, we believe the answer is yes, since intuitively by modeling the nonstationarity in the process, we can gradually identify the variables layer by layer. However it is clear that this will involve a lot of nontrivial technical details which we are not completely sure, we will continue working on it in the future work.
>
> > Q4: What's the specific meaning of "disentangled" representation in the title as well as the model name?
>
> While there is no single formalized notion of disentanglement which is widely accepted, the key intuition is that a disentangled representation should separate the distinct, informative factors of variations in the data. We approach the disentanglement from the nonlinear Independent Component Analysis (ICA) point of view. In our definition, "disentangled representation" for time series data means the following things:
>
> 1. (Temporally conditional independence) Each dimension of learned $\mathbf{z}_t$ (i.e. $z\_{t,1}, z\_{t,2} \dots z\_{t,n}$) are conditional independent given the history $\mathbf{z}\_{\text{Hx}}$.
> 2. (Identifiability) The learned representation $\mathbf{z}_t$ consists with the ground truth $\mathbf{z}_t$ in the data generating process (up to component-wise transformation and permutation).
>
> The field ''disentangled representation learning'' has been widely/extensively studied in the past few years, but as pointed out by [1], existing works rely heavily on inductive bias and ''well-disentangled models seemingly cannot be identified without supervision''. That is saying even though existing models can disentangle different factors in an intuitive way, the factors/variables that the model discovered can be far from ground truth, which is not favorable. Finding such ground truth (identifiability) is a further important yet challenging task. In the nonlinear ICA literature, researchers established identifiability result to find the independent components in the data generating process, which is a well-disentangled model with theoretical guarantee.
>
> > Q5: And how the disentangled representation is achieved?
>
> NTDC achieved disentangled representation by performing nonlinear ICA on nonstationary time series data. Specifically, NTDC first learned the nonstationarity of the process and give estimation of the domain indices $c_t$, and then NTDC leveraged the learned $c_t$ to find the independent components $\mathbf{z}_t$. The identifiability result guarantee that (1) each dimension of the learned $\mathbf{z}_t$ are conditionally independent which enforces the disentanglement, and (2) the learned $\mathbf{z}_t$ are equivalent to the ground truth value up to permutation and component-wise transformation, which means that NTDC can really recover the truth.
>
> > There are some grammatical errors in the main paper.
>
> We thanks the reviewer for pointing out this, we have proofread the manuscript and will update it in the final version.
>
> > The authors provide their code with anonymity, but there is limited explanations of the codes that which is directly linked to the proposed NTDC and how one can run the code.
>
> We thanks the reviewer for raising this issue, the training python scripts `train_{exp name}.py` are in the root directory of the repository and the our proposed NTDC method is coded with name `hmmxtdrl`. A sample run script for simulation data is provided as follow: `python train_simulation.py -c configs/simulation/simulation_hmmxtdrl.yaml` and sample config files are provided in the `configs` folder.
>
> Please let us know if you have further concerns and we are happy to provide more detailed information in the discussion phase.
> # References
>
> [1] Locatello, Francesco, et al.
> ''Challenging common assumptions in the unsupervised learning of disentangled representations.'' international conference on machine learning. PMLR, 2019.

---

> > ### Author Response · Authors · 2023-08-13
> >
> > Thank you once again for your thoughtful comments on our paper. Should there be any further questions or concerns, please let us know and we stand ready and eager to address them. We highly value your insights and would be more than pleased to provide any additional information or clarification you may require.

---

> > ### Comment · Area_Chair_BrYY · 2023-08-18
> > **Thanks**
> >
> > Thank you for this feedback authors. This will be taken into account.

---

> > ### Comment · Reviewer_t3u7 · 2023-08-19
> > **increase score from 4 to 4.5**
> >
> > Thank you for your clarification, especially for Q3, Q4, and Q5. I've read the authors' feedback, as well as other reviewers' reviews and the corresponding rebuttal. I've increased my score to 4.5, which is a definite borderline. In the meantime, I've left the rating to 4 since there is no 4.5 option.

---

> > > ### Author Response · Authors · 2023-08-19
> > > **Thanks and please let us know if you have further concern**
> > >
> > > Dear Reviewer t3u7,
> > >
> > > Thanks for acknowledging our clarification during rebuttal, we are delighted that your concerns have been resolved. Do you have any further questions or concerns that prevent you recommending acceptance to our paper? Please let us know and we are eager to clarify any of your concerns.
> > >
> > > With best regards,
> > >
> > > Authors of submission 9100

---

> ### Author Response · Authors · 2023-08-17
> **Possible to provide your feedback soon so we can reply?**
>
> Dear Reviewer t3u7,
>
> Thanks for your time and comments! Hope we are not bothering you, but we are looking forward to seeing whether our response and revision properly address your concerns and whether you have any further concerns, to which we hope for the opportunity to respond.
>
> We hope you will consider this work as an essential step towards unsupervised deep learning and temporal disentanglement, especially in the scenario in which the nonstationarity or domain information is unknown.
>
> With best regards,
>
> Authors of submission 9100

---

### Author Response · Authors · 2023-08-15
**Thank you and we are looking forward to your continued feedback!**

Dear Area Chair and all Reviewers:

Thanks again for all the insightful comments and advice, which helped us improve the paper's quality and clarity.

The discussion phase has been on for several days and we have not heard post-rebuttal discussion yet.

We would love to convince you of the merits of the paper. Please do not hesitate to let us know if there are any additional information or clarification that we can offer to make the paper better. We appreciate your comments and advice.

Best,

Authors

---

### Author Response · Authors · 2023-08-19
**Thanks again and we are looking forward to your continued feedback**

Thank you very much again for the insightful comments. We have been dedicated to absorbing them and providing our responses accordingly. As Aug 21 is the last day for us to respond, we hope for the chance to see and respond to your feedback.

It seems that Reviewer 5DXD hasn't got time to discuss whether our response properly addresses the concerns. We are eager to see the feedback.

We summarized our response into following categories:

**Updated presentation**

1. We have provided additional motivational examples with explanations to demonstrate the use case for our proposed model structure (**t3u7**)
2. We clarified domain specific terminology and provided detailed explanation on unclear descriptions in the manuscript, we also updated the related work part (**5DXD**)
3. We also revise and improve the presentation (**eHJB**)

**Newly conducted experiments**

1. We conducted additional experiments on the baseline settings that reviewers suggested (**5DXD**) and the result is consistent with our original claim.
2. Some of the metric doesn't fit the experiment setting, either the algorithm doesn't have such output or the algorithm doesn't converge in that setting. We added detailed explanation in the discussion (**eHJB**)
3. Some reviewers suggest comparison with methods with orthogonal contribution directions (**5DXD**), we thank the reviewer and conducted the experiments, the result is also consistent with our original conclusion, i.e. it is necessary to gain identifiability in representation learning and only an identifiable model can recover the ground truth information.

**Clarification on the Assumptions**

1. Thanks the reviewer for pointing out, we further discuss when the assumptions hold and when the assumptions fail. We clarified the sufficient variability assumption with examples which is well received by reviewer (**PKrt**)

---

### Decision · Program_Chairs · 2023-09-21

**Decision:**

Accept (poster)

**Comment:**

The reviewers were split about this paper and did not come to a consensus: on one hand they appreciated the experimental results and the applicability of the graphical model for which the paper derives identifiability results; on the other they had doubts about (1) its assumptions, (2) comparisons with related work, and (3) the clarity of the writing. After going through the paper and the discussion I have decided to vote to accept because the authors respond convincingly to each of the main concerns. Specifically, for (1) the reviewers argued that the assumptions necessary for Theorem 2 (which derives identifiability for latent variables z), are unintuitive and do not seem testable, making the theorem useless. The authors responded by pointing out that this is equivalent to a sufficient variability assumption made in GCL (Hyvarinen et al., 2019) which is testable. This fully resolves the reviewers' concerns. For (2), the reviewers argued that the authors forgot to compare with the field of sequential disentanglement and should add these comparisons. The authors have added a comparison with one of the sequential disentanglement methods suggested by the reviewers SKD (Berman et al., 2023). Further, they point out that the goal of sequential disentanglement is different as those works are not concerned with identifiability. Finally, they also point out that SKD can be used as the encoder-decoder module within their algorthm. This response convinces me that their approach is complimentary to the sequential disentanglement literature, resolving the point. For (3), the reviewers argue that many points in the paper are unclear: i. the end of Section 4.2 is a half sentence (which we later find out from the authors is missing an entire aligned equation block); ii. grammatical errors; iii. some confusing statments in the abstract including “In other words, existing identifiability results for general sequential data cannot benefit from those dual advantages simultaneously”: the advantages here aren’t clearly identified, I assume the authors mean auxiliary variables and the Markov assumption, but if that’s true then their work doesn’t benefit from both of these advantages either, it completely disregards auxiliary variables. This is itself a good thing, so it’s particularly odd that these additional assumptions are described as “advantages”, to resolve this I would remove this sentence; iv. the definition of sufficient variability in Theorem 2, this needs to be clarified as suggested by reviewer PKrt; v. the Figure 1 caption, the mouse plots (e) and (f) need to be connected to the graphical model (d) in the caption, similar to how this is done in the text. Ultimately, paper clarity (3) is the most serious concern of the paper. However, the authors have convinced me that they are able to fix these concerns in the final version, based on their responses to reviewers. For these reasons I argue for acceptance. Authors: please make sure to carefully incorporate all of the reviewer concerns, particularly w.r.t. clarity into the final version. After you have done so, I strongly recommend you pass through the paper multiple times more to make sure there are no mistakes or omissions, and that the paper has clear examples to demonstrate the most intricate concepts. Once this is done, the paper will make a great contribution to the conference!